# Pyroelectric nanoplates for reduction of $CO_2$ to methanol driven by temperature-variation

Lingbo Xiao[1,5], Xiaoli Xu[1,5], Yanmin Jia[2✉], Ge Hu[3], Jun Hu [3✉], Biao Yuan [4], Yi Yu [4] & Guifu Zou [1✉]

Carbon dioxide ($CO_2$) is a problematic greenhouse gas, although its conversion to alternative fuels represents a promising approach to limit its long-term effects. Here, pyroelectric nanostructured materials are shown to utilize temperature-variations and to reduce $CO_2$ for methanol. Layered perovskite bismuth tungstate nanoplates harvest heat energy from temperature-variation, driving pyroelectric catalytic $CO_2$ reduction for methanol at temperatures between 15 °C and 70 °C. The methanol yield can be as high as 55.0 $\mu mol \cdot g^{-1}$ after experiencing 20 cycles of temperature-variation. This efficient, cost-effective, and environmental-friendly pyroelectric catalytic $CO_2$ reduction route provides an avenue towards utilizing natural diurnal temperature-variation for future methanol economy.

[1] College of Energy, Soochow Institute for Energy and Materials Innovations, and Key Laboratory of Advanced Carbon Materials and Wearable Energy Technologies of Jiangsu Province Soochow University, 215006 Suzhou, China. [2] School of Science, Xi'an University of Posts & Telecommunications, 710121 Xi'an, China. [3] School of Physical Science and Technology & Jiangsu Key Laboratory of Thin Films, Soochow University, 215006 Suzhou, China. [4] School of Physical Science and Technology, ShanghaiTech University, 201210 Shanghai, China. [5]These authors contributed equally: Lingbo Xiao, Xiaoli Xu. ✉email: jiayanmin@xupt.edu.cn; jhu@suda.edu.cn; zouguifu@suda.edu.cn

For hundreds of years, fossil fuels have been the main energy source for human activities and industrial manufacture. With the development of human society, the decrease in fossil energy and the increase of $CO_2$ concentration have aroused great attention. For instance, energy crisis, greenhouse effect and ocean acidification are some of the main problems facing humanity[1–3]. Converting $CO_2$ into hydrocarbon fuels is considered as one of the ideal solutions, which can solve not only the environmental problems but also the high requirements of energy consumption. Various methods have been explored to convert $CO_2$ to organic fuels, such as photocatalytic reduction, electrocatalytic reduction, biological transformation, hydrogenation, and dry reforming[4–7]. Nevertheless, hydrogenation of $CO_2$ to form $CH_3OH$ process requires high operating temperatures (200–250 °C) and high pressures (5–10 MPa), which limit the yield of methanol[8]. As a matter of fact, photocatalytic reduction of $CO_2$ can be carried out at mild temperature and pressure, but it does not work in dark[9].

Temperature variation is a recurring phenomenon in our daily life[10]. It would be meaningful to harvest such abundant energy source during temperature variation. Such a motive is reasonable because pyroelectric materials can convert heat energy into electric energy via repeating cooling or heating process[11–13]. Pyroelectric materials can produce positive and negative electric charges during temperature variation. The free charges generated through pyroelectric process can be applied to catalytic processes such as dye decomposition[14–16] and water splitting[17,18]. Theoretical calculation shows that a pyroelectric engine in an ideal condition can reach an energy conversion efficiency as high as 84–92%, which is much higher than the photovoltaic energy conversion efficiency typically in the range of 20%[19,20]. Theoretically, Kakekhani et al.[21] have proved the feasibility of pyroelectric catalytic water splitting. However, to our best knowledge, there is no report about collecting the energy using pyroelectric materials from temperature variation for $CO_2$ reduction.

The catalytic performance of ferroelectrics has been studied for 70 years. For example, the internal fields from the polarization of the ferroelectrics can separate electrons and holes, thus enhancing the catalytic efficiency[17]. Ferroelectric polarization can affect molecular adsorption and desorption from the surface of the materials[21]. It is well known that all ferroelectric materials are pyroelectric materials. As the simplest member of bismuth layer-structured Aurivillius phase, bismuth tungstate ($Bi_2WO_6$) exhibits good ferroelectric and pyroelectric properties. Meanwhile, $Bi_2WO_6$ has some other interesting properties such as high ion conductivity, large spontaneous polarization ($P \cong 50\ \mu C\ cm^{-2}$), high Curie temperature ($T_C \cong 950$ °C), and photocatalytic property[22,23]. As $Bi_2WO_6$ is constructed by alternating $(Bi_2O_2)^{2+}$ and $(WO_4)^{2-}$ layers, such a layered structure enables high thermal and chemical stabilities[24]. More importantly, the suitable energy band structure and surface properties of $Bi_2WO_6$ allow it for $CO_2$ reduction into renewable hydrocarbon fuel[25,26]. In this work, through pyroelectric catalysis, $CO_2$ is reduced to $CH_3OH$ at temperature variation below 100 °C. The efficiency has reached as high as 55.0 $\mu mol\ g^{-1}$ after experiencing 20 cycles between 15 °C and 70 °C. Our experimental work provides a new route to $CO_2$ reduction for methanol through a pyroelectric catalytic process, which can be carried out near room temperature.

## Results and discussion

**Characterization of $Bi_2WO_6$.** Previous study shows that $Bi_2WO_6$ is ferroelectric with an orthorhombic structure[27]. $Bi_2WO_6$ nanoplates were synthesized by hydrothermal process in this work (see the experimental details in the section of Methods). In order to identify the phase of the synthesized $Bi_2WO_6$, X-ray diffraction (XRD) analysis was performed at room temperature. As shown in Fig. 1a, all the diffraction peaks can be assigned to $Bi_2WO_6$ according to the standard JCPDS card No. 79-2381 (space group: $Pca2_1$; point group: $mm2$; orthorhombic crystal system).

It can be seen from Fig. 1b that the synthesized $Bi_2WO_6$ has a plate-like morphology with an average size of 250 nm. Figure 1c presents the image of $Bi_2WO_6$ from transmission electron microscopy (TEM), where the nanoplate has the similar feature size as the ones of scanning electron microscope (SEM) images in Fig. 1b. The high-resolution transmission electron microscopy (HRTEM) image of $Bi_2WO_6$ is shown in Fig. 1d. It clearly shows the single-crystalline nature of $Bi_2WO_6$ nanoplate with a lattice plane intervals of 0.27 nm, corresponding to the (002)/(200) plane, respectively. The aberration-corrected high-angle annular dark field scanning transmission electron microscopy (HAADF-STEM) image of $Bi_2WO_6$ sample is presented in Fig. 1e. The light/dark gray contrast spots correspond to Bi ($Z = 83$, where $Z$ is the atomic number) and W ($Z = 74$) atom columns, respectively. The inset image denotes the fast Fourier transformation (FFT) of the STEM image. The FFT image shows the zone axis of the STEM image is [103], which is perpendicular to $b$ direction. Therefore, the STEM image reflects the layered structure of $Bi_2WO_6$, which is sandwiched by alternating perovskite-like $(WO_4)^{2-}$ and fluorite-like $(Bi_2O_2)^{2+}$ blocks. A comparison between the STEM image and the structure model is schematically illustrated in Fig. 1f. The left picture in Fig. 1f is the magnified image of the area marked in a red rectangle in Fig. 1e. The inset in Fig. 1e shows the simulated diffraction pattern in the [103] projection direction. Complete structure model is shown on the right side of Fig. 1f.

$Bi_2WO_6$ is paraelectric with a high-symmetry body-centered tetragonal structure (space group symmetry $I4/mmm$) at high temperature. When the temperature drops, symmetry of the crystal structure will be broken, and the distortion of the symmetry tetragonal structure makes $Bi_2WO_6$ generate ferroelectric properties. This mainly includes two aspects. First, the ions displace along the [110] axis of the tetragonal structure. Secondly, the $WO_6$ octahedra rotates around the $a$ and $c$ axes[28]. In order to characterize the ferroelectric properties of the as-synthesized $Bi_2WO_6$ nanoplates, ferroelectric domains of $Bi_2WO_6$ nanoplates are observed using a piezoelectric force microscope (PFM) at a slow scanning frequency of 1 Hz with an area of $0.8 \times 0.8\ \mu m^2$. The nanoplates' morphologies of $Bi_2WO_6$ in Fig. 2a are consistent with the results of TEM and SEM. Figure 2b, c show the vertical piezoresponse amplitude and phase image, respectively. The distinct contrast in the images illustrates the different polarization in the $Bi_2WO_6$ nanoplates. Figure 2d, e display the local piezoelectric hysteresis loops of the $Bi_2WO_6$, including both "off" state (piezoelectric displacement contribution only) and "on" state (both the piezoelectric contribution and the displacement resulted from electrostatic interaction). The phase angles at "off" and "on" states change about 150° under 60 V DC bias field, confirming the occurrence of a local polarization switching under an electric field. The butterfly-shaped hysteresis loop further verifies the local ferro-/piezoelectric response of $Bi_2WO_6$ nanoplate.

It is noted that $Bi_2WO_6$ also shows pyroelectric properties, where imbalanced polarization charges can generate electric field when the material undergoes temperature variation. The voltage produced by pyroelectric effect can be driving force for electrochemical reactions. Figure S1 shows the pyro-potential distribution across a $Bi_2WO_6$ nanoplate fitted by COMSOL finite element simulation, in which different colors represent different potentials. It can be seen that potential difference occurs on the surfaces of the $Bi_2WO_6$ nanoplates. In general, ferroelectric

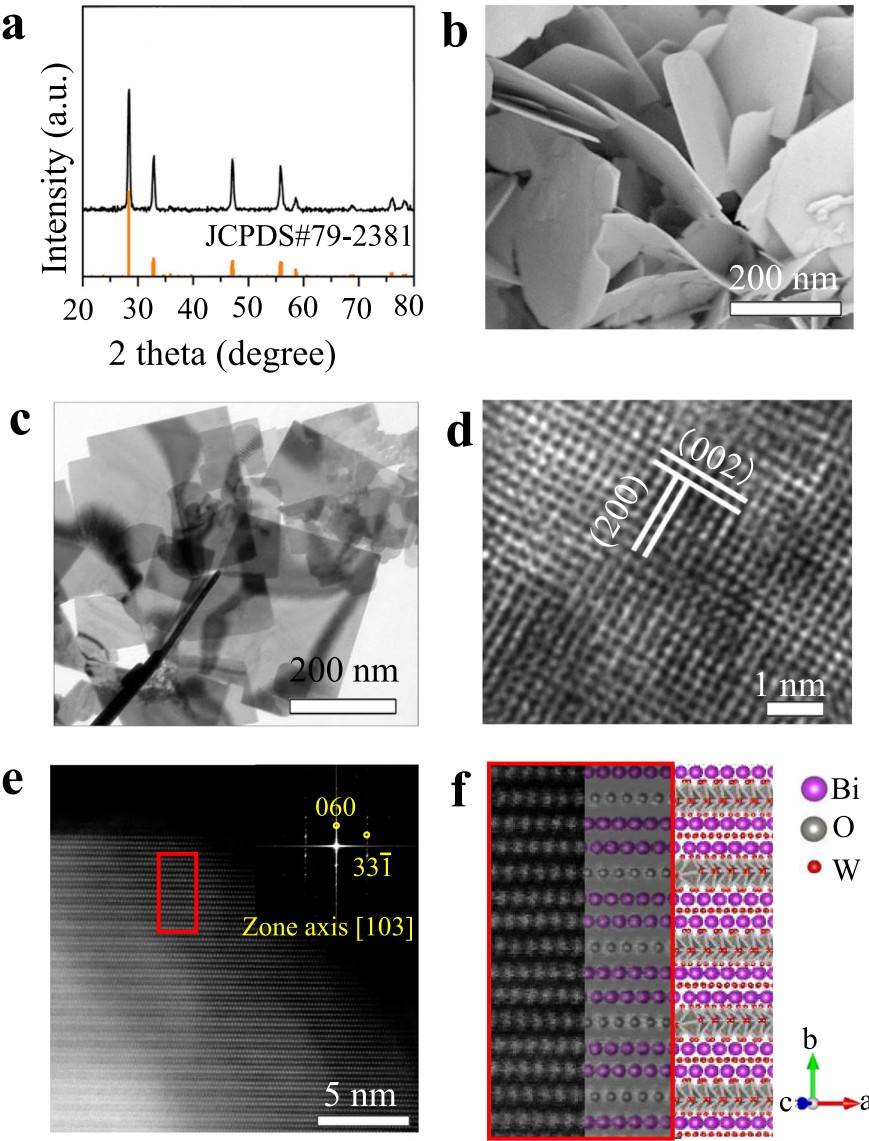

**Fig. 1 Structures of Bi₂WO₆ sample. a** XRD pattern, **b** SEM photograph, **c** TEM photograph, **d** HRTEM image, and **e** Aberration-Corrected HAADF-STEM image. The inset image in **e** denotes the FFT of STEM. **f** Comparison between the STEM image and the structure model of the layered structure of Bi₂WO₆.

materials have greater pyroelectric and piezoelectric coefficients than non-ferroelectrics[28]. In order to demonstrate $Bi_2WO_6$ generate free charges through temperature variation, pyro-current response of $Bi_2WO_6$ nanoplates is measured. Once the temperature of the pyroelectric material changes, the pyroelectric charges can be generated quickly due to the pyroelectric effect. The pyroelectric current can be expressed by Eq. (1),

$$I_{pyro} = p \cdot A \cdot (dT/dt) \tag{1}$$

where $I$ and $p$ are the pyroelectric current and the pyroelectric coefficient of $Bi_2WO_6$, respectively. $A$ is the area of electrode. $dT/dt$ is the rate of temperature fluctuation. Therefore, the pyroelectric current is proportional to $dT/dt$, any temperature change of the pyroelectric material will cause it to generate free charges. Figure 2f, g show the current change generated by $Bi_2WO_6$ nanoplates with the infrared signal. Under infrared radiation, a sharp increase of current density is induced by the pyroelectric effect due to the rapid increase of temperature within $Bi_2WO_6$ nanoplates. The current density decays slowly due to the decrease of temperature change rate, and maintains at a steady value under the equilibrium condition. When the infrared radiation is turned off, due to the

instantaneous temperature decrease ($dT/dt < 0$), the redistribution of the pyroelectric charges will produce a reverse current. The output current returns to zero while there is no temperature change and infrared radiation. To clarify the temperature effect, we further use xenon lamp (UV light) instead of the infrared radiation to illuminate the sample. There is no pyro-current signal generate, which further confirms that $Bi_2WO_6$ generates free charge under temperature variation(see Fig. S2).

**Pyroelectric catalytic activity of Bi₂WO₆.** In order to evaluate the pyroelectric catalytic activity of $Bi_2WO_6$, $CO_2$ reduction test was carried out under the condition of temperature variation. As shown in Fig. 3a, the methanol yield increases with the increasing thermal cycles. The total methanol yield reaches 20.5 µmol g⁻¹ without adding any sacrificial agent after 20 thermal cycles. ¹H NMR in Fig. S3 also demonstrates that no other products can be detected in the liquid phase. Meanwhile, the analyses of gaseous products in Fig. S4 show only a small amount of $CH_4$ and CO (0.11 µmol g⁻¹ and 0.20 µmol g⁻¹, respectively), indicating high selectivity of $Bi_2WO_6$ pyroelectric catalytic $CO_2$ reduction to $CH_3OH$. The oxygen production detection is shown in Fig. S5,

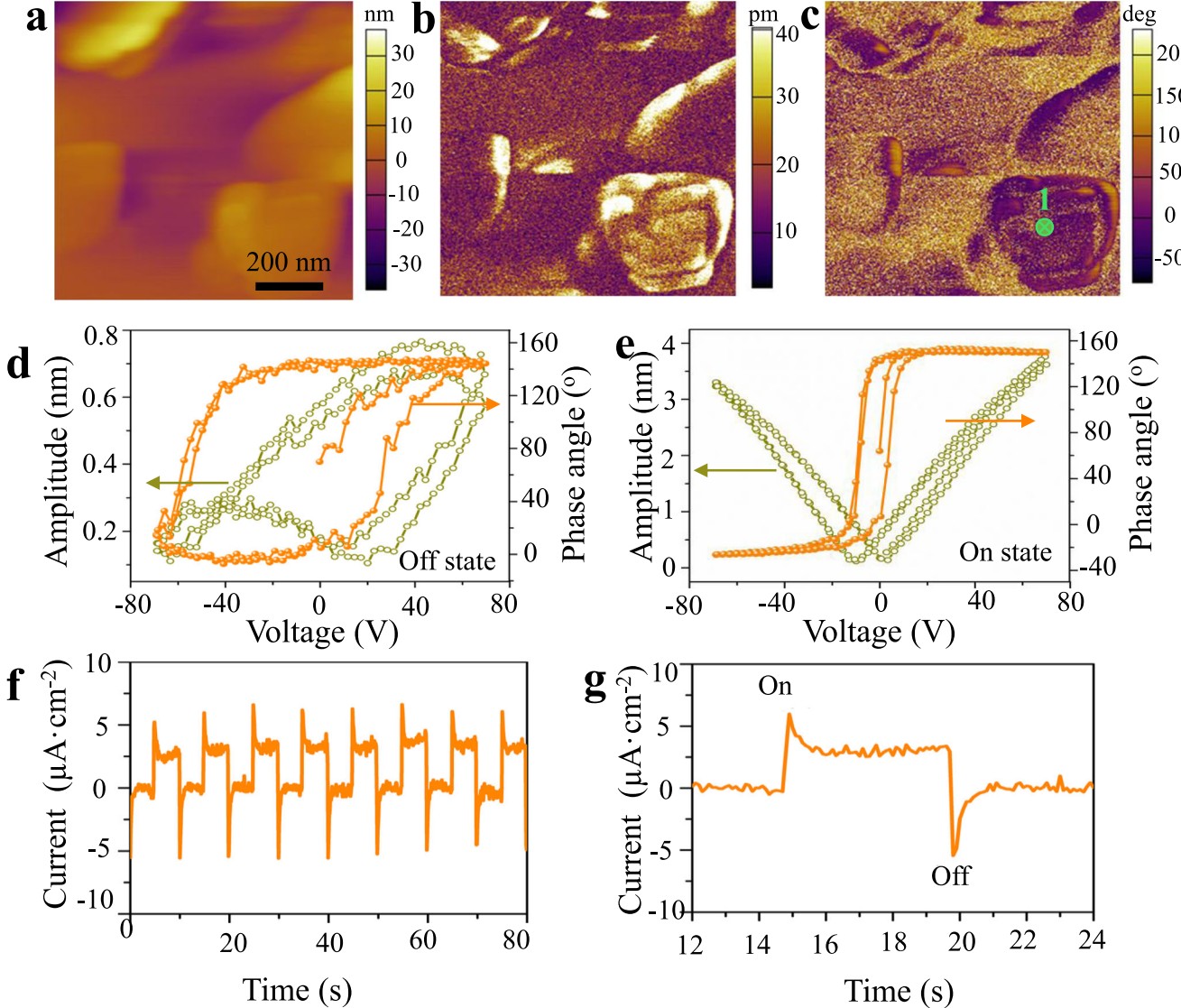

**Fig. 2 Ferro-/pyroelectric properties of $Bi_2WO_6$. a** Topology, **b** vertical amplitude, and **c** phase images of $Bi_2WO_6$ nanoplate. The local hysteresis loops of $Bi_2WO_6$ nanoplate for point 1 marked in **c**: **d** "off" state, and **e** "on" state. **f** The Pyro-current response of $Bi_2WO_6$, **g** enlarged view of one full light on/off cycle is shown in **f**.

the amount of $O_2$ is roughly 1.5 times the amount of methanol. It has been similarly reported that the photocatalytic process works due to the recombination of electrons and holes. Such a process will significantly affect the catalytic efficiency[29,30]. To reduce the occurrence of electrons recombination with holes, sacrificial agents are usually added to the reaction system. Figure 3b shows that more methanol can be generated by using $Na_2SO_3$ as negative charge sacrificial agent. The methanol yield can be as high as 55.0 μmol $g^{-1}$ after 20 temperature-variation cycles, which is 2.5 times more than that without $Na_2SO_3$. The range of temperature variation can also affect the pyroelectric catalytic $CO_2$ reduction. Figure S6 shows the yield of methanol after 10 thermal cycles in different temperature ranges (15–40 °C, 15–50 °C, 15–70 °C, 15–85 °C). In Fig. S6, the methanol yield increases as the temperature range increases. The pyro-induced charges (dQ) can be expressed in Eq. (2)

$$dQ = p \cdot A \cdot dT \tag{2}$$

A larger temperature range can generate more pyro-charges, leading to better pyroelectric catalytic results. It is also noted that

the $Bi_2WO_6$ nanoplates maintain their crystal structure and morphology after pyroelectric catalytic reduction as confirmed by the XRD analysis and SEM characterization (see Fig. S7). To further prove that $CO_2$ reduction comes from pyroelectric catalysis, experiment without $Bi_2WO_6$ nanoplates is performed, the result in Fig. S8a shows that no methanol or other products can be detected under temperature variation without $Bi_2WO_6$ nanoplates. Furthermore, no methanol or other products can be detected when the test is carried out with the presence of $Bi_2WO_6$ nanoplates for 10 h at temperatures of 15 °C, 45 °C and 70 °C, respectively (Fig. S8b). $CO_2$ is a linear molecule, which is one of the most thermodynamically stable carbon compounds, it's hard to break the bonding of C=O[8,31].

Pyroelectric charges can react with $O_2$ and $OH^-$ in water to form $O_2^{·-}$ and $·OH$. Such reactions can be expressed as shown in Eqs. (3)–(5).

$$Bi_2WO_6 \xrightarrow{\Delta T} Bi_2WO_6(q^+ + q^-) \tag{3}$$

$$O_2 + q^- \rightarrow O_2^{·-} \tag{4}$$

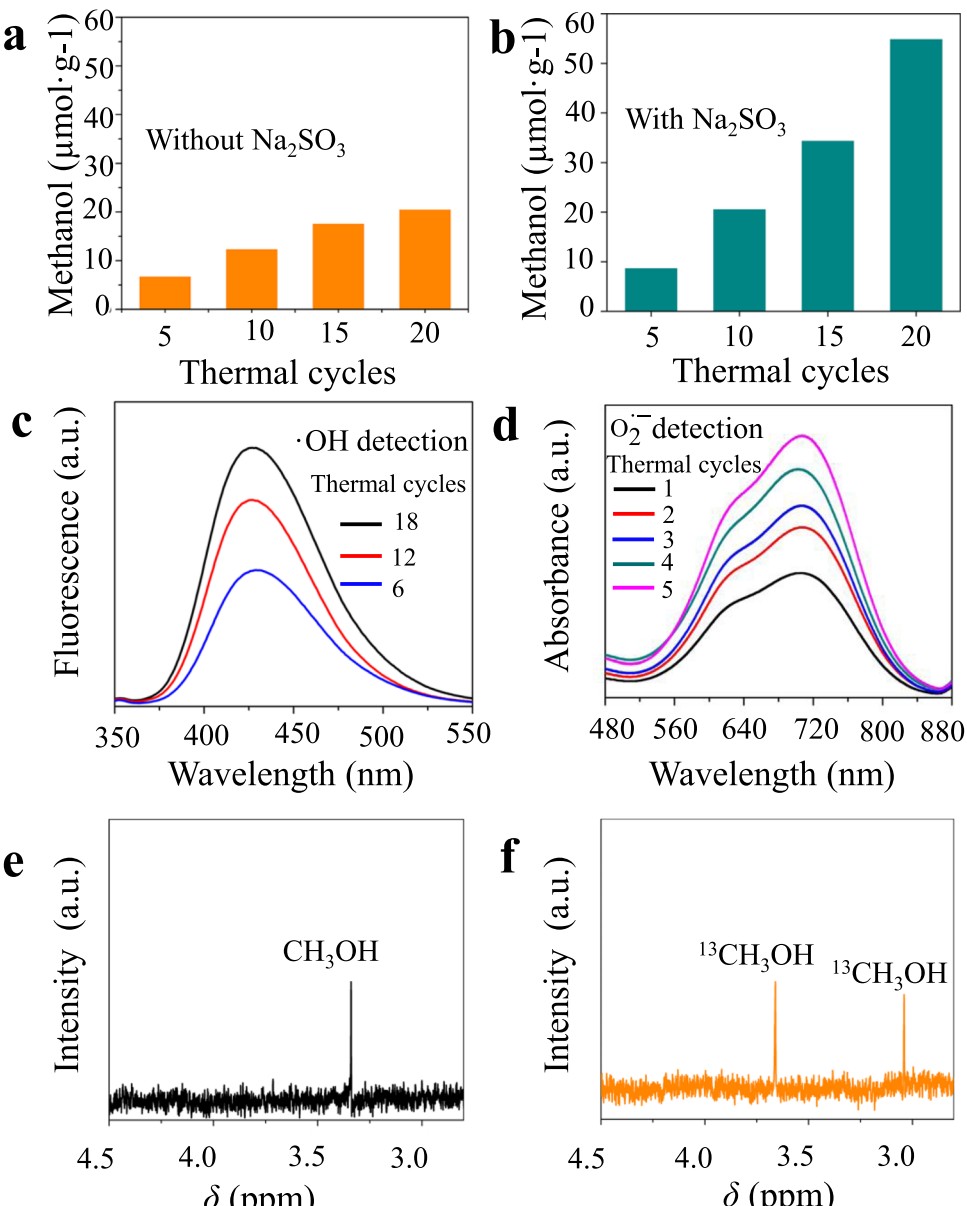

**Fig. 3 Catalytic activities of Bi₂WO₆. a** Methanol yield through pyroelectric catalytic CO₂ conversion without Na₂SO₃, and **b** with Na₂SO₃ as sacrificial agent. **c** The fluorescence spectra of 2-hydroxyterephthalic acid, **d** the absorption spectra of diformazan and monoformazan. **e** 1H NMR spectra of the pyroelectric catalytic reaction solution with unlabeled CO₂ and **f** 1H NMR spectra of the pyroelectric catalytic reaction solution with labeled ¹³CO₂.

$$OH^- + q^+ \rightarrow \cdot OH \qquad (5)$$

To have a better understanding of the pyroelectric catalysis, $O_2^-$ and $\cdot OH$ detections are performed. Experimentally, the $\cdot OH$ can be detected by fluorescence spectrometry using terephthalic acid as a photoluminescent $\cdot OH$ trapping agent. The $O_2^-$ can be detected by UV–Vis spectrophotometer since $O_2^-$ can react with nitro-blue tetrazolium (BNT) to produce diformazan and monoformazan. As shown in Fig. 3c, significant fluorescence emission at ~425 nm associated with 2-hydroxyterephthalic acid is observed upon the temperature-variation cycles. The gradual increase of luminescence intensity with temperature-variation cycles indicates the formation of $\cdot OH$. Figure 3d presents the absorption spectra of diformazan and monoformazan, which are produced by BNT reacted with $O_2^-$. The increase of peak absorption at ~630 nm and 720 nm with the temperature-variation cycles indicates the formation of $O_2^-$[32]. The electron

spin resonance characterization in Fig. S9 can further prove $O_2^-$ and $\cdot OH$ generated through temperature variation. $O_2^-$ and $\cdot OH$ are considered to be the main active species in dye decomposition[33]. Except for the pyroelectric catalysis of CO₂ reduction, we further performed RhB pyroelectric catalytic decomposition experiment to fully demonstrate the pyroelectric catalytic activities of Bi₂WO₆. In fact, RhB pyroelectric catalytic decomposition is a visual evidence to prove the redox ability of pyroelectric charges. Accordingly, Rhodamine B (RhB) solution (5 mg L⁻¹) is used to demonstrate the pyroelectric catalytic dye decomposition of Bi₂WO₆ in Fig. S10a, b. In order to prove that methanol is the product of CO₂ reduction, we manage the isotopic labeling experiment using ¹³CO₂ as feedstock. To avoid ¹²CO₂, we have done the additional experimental using NaOH instead of NaHCO₃. The ¹H NMR spectrum of the reaction solution (Fig. 3e) clearly shows the formation of methanol ($\delta$ = 3.34 ppm) when the unlabeled CO₂ is used as feedstock. While

using $^{13}CO_2$ instead of $CO_2$, the $^1H$ NMR spectrum of the reaction solution in Fig. 3f shows doublet peaks between 3.7 and 3.0 ppm, which is attributed to the proton coupled with the $^{13}C$ of $^{13}CH_3OH$[34]. The results directly indicate that $CO_2$ is the carbon source for the pyroelectric catalytic $CO_2$ reduction into $CH_3OH$. The time course change of the intensity is shown in Fig. S11.

**Theoretical calculation of $CO_2$ reduction reaction path.** To better illustrate the reaction mechanism for the $CO_2$ reduction, we employ first-principles calculations with SIESTA package, which is based on density functional theory (DFT)[35]. The pseudopotentials are constructed by the Troullier-Martins scheme[36]. The Ceperley–Alder exchange-correlation functional as parameterized by Perdew and Zunger is employed for the local density approximation (LDA)[37,38]. In all calculations, the double-ζ plus polarization basis sets are chosen for all atoms. The atomic structures are fully relaxed using the conjugated gradient method until the Hellman–Feynman force on each atom is smaller than 0.02 eV Å$^{-1}$. Since $Bi_2WO_6$ consists of alternative $(WO_4)^{2-}$ and $(Bi_2O_2)^{2+}$ layers as discussed above, a slab model is constructed for the $Bi_2WO_6(001)$ surface. The top of the slab is terminated by the WO square network, and the bottom of the slab is saturated by H atoms, as shown in Fig. S12. It is known that oxygen vacancies commonly exist in oxide semiconductors[39]. However, it is found that the formation energy of an oxygen vacancy in $Bi_2WO_6(001)$ is as large as 3.2 eV, which indicates that the density of oxygen vacancies in $Bi_2WO_6(001)$ is ignorable. To explore the possible process of the $CO_2$ reduction, it is necessary to figure out the ground-state adsorption configuration of the $CO_2$ molecule on $Bi_2WO_6(001)$. Figure 4a shows five different adsorption configurations for $CO_2$. The lowest adsorption energy is −3.6 eV, which implies that the $CO_2$ molecule is strongly bound to $Bi_2WO_6(001)$. In this case, the $CO_2$ molecule is bent with one C–W bond (2.00 Å) and two O–W bonds (2.09 and 2.26 Å) as shown in Fig. 4b (step "0"), which is different from previous report[9]. The C–O bond lengths are elongated by 0.1 Å because of the interaction between $CO_2$ and $Bi_2WO_6$ (001). The $CO_2$ reduction starts when the hydrogen ions in the solvent interact

with the $CO_2$ molecule. Note that DFT calculations, the hydrogen atom as proton (H$^+$) and electron (e$^-$) cannot be separated directly. To model the reaction between H and radical on $Bi_2WO_6(001)$, a H atom is placed beside a certain site of the radical and carried out DFT calculations to optimize the interaction between them. Electron charge transfer happens between the H atom and radical, usually from H to the radical so that the H atom finally becomes H$^+$, according to the chemical bonding between H and the radical (e.g., $CO_2$ molecule in this work). In other words, charge separation can be reached after self-consistent-field iterations. In addition, the gas phase of H is assumed because the solvent does not involve in the reaction of the $CO_2$ reduction. The process of the $CO_2$ reduction is divided into a series of steps and the reaction energies are calculated step by step. All possible structural configurations along with addition of one H ion are considered for each step simulation. For instance, from step "0" to step "1", the H ion may bind to the $CO_2$ molecule through C or O atom, or to the $Bi_2WO_6$ (001) surface through W or O atoms, which results in different products. To determine the most possible reaction, the reaction energy of each product is estimated as: $\Delta E = E(n) - E(n-1) - \mu_H$, where $E(n)$ is the total energy of a certain product at the $n^{th}$ step and $\mu_H$ is the chemical potential of H. After optimizing all structural configurations, the case with the lowest reaction energy at each step is plotted in Fig. 4b. Obviously, the structural configurations in Fig. 4b are the most possible products for each step. Further, Fig. 4c shows the reaction energy of the most possible product at each step. Here, the first three H ions at the first three steps prefer to bind to the C atom, and these reactions are exothermic due to the large negative reaction energies as seen in Fig. 4c. As a consequence, one C–O bond is broken, then a $CH_3O^*$ radical and a separate O ion are produced at step "3". Then the subsequent H ions will be attracted by the separated O ion until a $H_2O$ molecule forms (step "5"). However, the $H_2O$ molecule is not released from $Bi_2WO_6$ (001), because it requires a large activation energy of about 1.7 eV. Finally, a methanol ($CH_3OH$) molecule is produced after one more H ion attaches to the $CH_3O^*$ radical (step "6"). As shown in Fig. 4c, the first four reaction steps are exothermic while the last three are endothermic. In particular, an activation energy

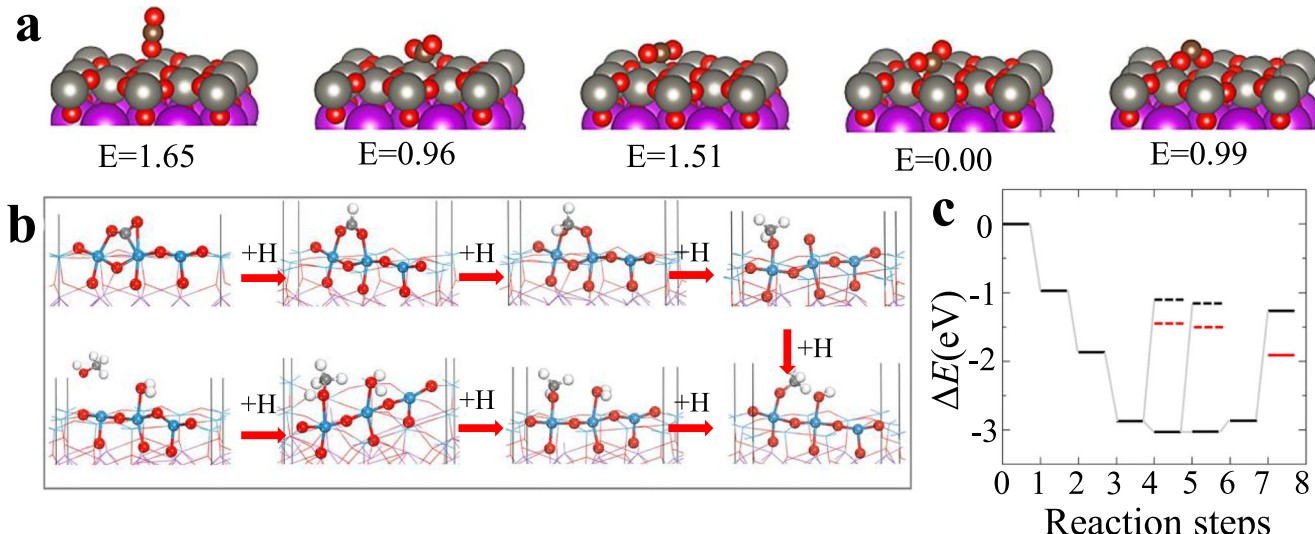

**Fig. 4 Adsorption configurations and reaction path of $CO_2$ into $CH_3OH$ on $Bi_2WO_6$. a** Five different adsorption configurations for $CO_2$ on $Bi_2WO_6$. The red, gray, purple, and brown spheres stands for O, W, Bi, and C atoms. **b** Structures and **c** reaction energies for the $CO_2$ reduction. Eight reaction steps are considered. The red, cyan, light gray, and dark gray spheres stand for O, W, H, and C atoms. Only the atoms around $CO_2$ are highlighted. The reaction energies for the side product (methanol) at steps "4" and "5" are indicated by short dashed lines. The short red (dashed) lines are estimated from the free energies.

of 1.6 eV (equal to the reaction energy) is needed for the $CH_3OH$ molecule to be detached from the $Bi_2WO_6(001)$ surface, i.e., from step "6" to step "7". Note that $CH_3OH$ may also be produced at steps "4" and "5" (short dashed lines in Fig. 4c), but the corresponding activation energies are 1.77 and 1.87 eV, respectively, even larger than that for step "7". Therefore, the possibility to produce $CH_3OH$ at steps "4" and "5" is very small, because the major reactions at the first two steps are exothermic. Nevertheless, the overall process of the $CO_2$ reduction is still exothermic as shown in Fig. 4c, that is, the activation energies in the last three reaction steps can be compensated by the energy released in the first four steps. In principle, the $CO_2$ reduction happens spontaneously. However, energy supply might be required while the energy loss in a solvent environment. This is the reason that the temperature is not high in our experiments. It is worth pointing out that, when the $CH_3OH$ molecule is detached from the $Bi_2WO_6$ (001) surface, the zero-point energy and enthalpy contribute to the free energy significantly[40]. Therefore, the reaction energies are also estimated through the free energies for $CH_3OH$ production indicated by the red lines in Fig. 4c. Interestingly, the activation energy of step "7" decreases to 0.95 eV, which implies that the reaction may happen at relatively high temperature, as found in our experiments.

On the basis of the above analysis, the mechanism of pyroelectric catalytic $CO_2$ reduction induced by temperature variation is proposed as shown in Fig. 5. When the temperature of $Bi_2WO_6$ remains stable, the internal spontaneous polarization is balanced with the external bound charges (Fig. 5a). It has been reported that the spontaneous polarization intensity of pyroelectric materials decreases as its temperature increases[41]. That is to say, the increase in temperature will reduce the polarization of the pyroelectric catalyst, thereby breaking the balance and generating free charges. The free negative charges react with adsorbed $CO_2$ to form methanol and the free positive charges would be captured by $Na_2SO_3$ to form $Na_2SO_4$ (Fig. 5b). As a result, balance is established again between polarization and bonding charges (Fig. 5c). On the other hand, the decrease of temperature causes the increase of spontaneous polarization, and the equilibrium will be broken again, thus leading to opposite charges transfer and $CO_2$ reduction process (Fig. 5d). Then the catalyst temperature returns to its original value and at the same time it returns to its original equilibrium. Therefore, the continuous thermal cycles will cause continuous $CO_2$ reduction reaction.

In summary, this work introduces the use of pyroelectric materials to reduce $CO_2$ to methanol driven by temperature variation. Experimental results show that the yield of methanol

generation from $CO_2$ can be as high as 55.0 $\mu mol\,g^{-1}$ after 20 cycles of temperature variation. This efficient and environmentally friendly process based on the pyroelectric nanomaterial $Bi_2WO_6$ provides great potential for $CO_2$ reduction in utilizing environmental heat energy near room temperature.

## Methods

**Materials**. All used chemicals are analytic grade reagents without further purification. Bismuth nitrate ($Bi(NO_3)_3\cdot5H_2O$, AR), sodium tungstate ($Na_2WO_4\cdot2H_2O$, AR) and sodium bicarbonate ($NaHCO_3$, AR), dimethyl sulfoxide (DMSO, AR) were acquired from Sinopharm Chemical Reagent Co., Ltd. Sodium sulfite ($Na_2SO_3$, AR was purchased from Shanghai Macklin Biochemical Co., Ltd. Carban dioxide ($CO_2$, ≥99.995%) was purchased from Soochow Jinhong Co., Ltd. Deuterium oxide ($D_2O$, ≥99.9%) was purchased from Qingdao Asfirst Science and trade Co., Ltd. $^{13}CO_2$ was bought from Soochow changyou gas Co., Ltd with purity of 99.9%. Deionized water was employed throughout the whole experiments.

**Preparation of $Bi_2WO_6$ nanoplates**. $Bi_2WO_6$ nanoplates were synthesized through hydrothermal process. In a typical process, 485 mg of $Bi(NO_3)_3\cdot5H_2O$ (1 mmol) and 165 mg of $Na_2WO_4\cdot2H_2O$ (0.5 mmol) were added into the mixed solution. White precipitate appeared immediately in the solution. After being washed for several times, the collected precipitate was added into a 50 mL Teflon-lined autoclave and filled with deionized water up to 80% of the total volume. Then the autoclave was sealed into a stainless steel tank and kept at 433 K for 20 h. After reactions, the white as-prepared sample was centrifuged at $2400 \times g$ and washed three times with deionized water. Finally, the collected products were dried in vacuum at 333 K for 12 h for further use.

**Characterization**. The crystal structure was test by an X-ray diffractomer (Philips PW3040/60, the Netherlands) with monochromatic Cu Kα radiation (λ = 1.5406 Å, $2\theta = 20°-80°$). The morphologies of the $Bi_2WO_6$ sample was characterized by a transmission electron microscope (TEM, Hitachi H-7650, Japan) and a field emission transmission electron microscopy (FETEM, Scios, USA) with an accelerated voltage of 200 kV. The high-resolution transmission electron microscopy (HRTEM) image was acquired through a field emission transmission electron microscopy (FEI Tecnai G2 F20 S-TWIN, USA) with the accelerated voltage of 200 kV. Aberration-corrected high-angle annular dark field scanning transmission electron microscopy image was obtained on a 300 kV aberration-corrected JEM-ARM300F. The piezoelectric property of the $Bi_2WO_6$ sample was characterized with piezoresponse force microscopy (PFM, MFP-3D, USA). Photoluminescence (PL) measurements were carried out with a Horiba spectrofluorometer (Fluoromax-4, France) in air. Pyro-current response was measured on a CHI 660E electrochemical workstation using a three-electrode cell. The UV–visible absorption spectra are recorded on a UV2501PC (Shimadzu, Japan).

**Pyroelectric catalytic $CO_2$ reduction activity test**. In the pyroelectric catalytic $CO_2$ conversion process, $Bi_2WO_6$ powder (40 mg) was suspended in 5 mL 0.2 M $NaHCO_3$ solution in a 50 mL flask with the addition of 0.3 M $Na_2SO_3$ as sacrificial donor. High purity $CO_2$ gas was bubbled into the flask for 10 min. Then the flask was immediately sealed with a rubber stopper. Then the flask was immediately sealed with a rubber stopper. The sample was suspended in the solution under magnetic stirring, being applied alternating temperature between 15 °C and 70 °C in water bath. The entire catalytic process is performed in dark. The detailed

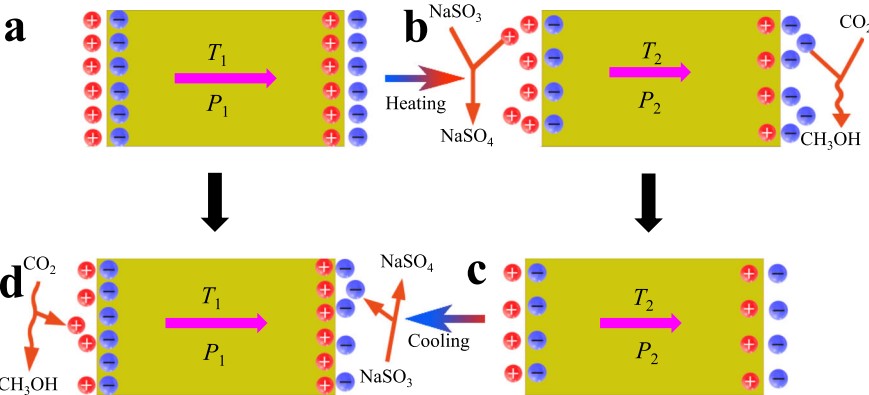

**Fig. 5 The mechanism of pyro-catalytic $CO_2$ reduction induced by pyroelectric $Bi_2WO_6$ nanoplate. a** The temperature of catalyst remains constant, its spontaneous polarization in equilibrium with the bound charges. **b** The rapid rise in temperature broke the balance, and thus induce $CO_2$ reduction reaction. **c** A new balance is established after the $CO_2$ reduction reaction. **d** When the temperature drops, the opposite charges transfer, leading to a new $CO_2$ reduction process.

temperature profile and schematic diagram of the process can be found from Figs. S13 and S14, respectively. To detect the formation of methanol, 1 mL solution was fetched out and analyzed by using a gas chromatograph (Persee G5) equipped with a KB-5 column connected to a flame ionization detector. For the nuclear magnetic resonance (NMR) test, 800 µL reaction solution, 100 µL $D_2O$ and 50 µL DMSO (0.1% vol aqueous solution) were taken into nuclear magnetic tubes, and detected with an NMR spectrometer with superconducting magnet (AVANCE NEO 400 MHz, Switzerland).

## Data availability

The data that support the findings of this study are available from the corresponding author upon reasonable request.

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

## Acknowledgements

We gratefully acknowledge the support from the National Natural Science Foundation of China (21971172, 21671141, 11574223, 51872264), the Natural Science Foundation of Jiangsu Province (BK20150303), the Priority Academic Program Development (PAPD) of Jiangsu Higher Education Institutions for Optical Engineering, Jiangsu Collaborative Innovation Center of Photovoltaic Science and Engineering, Shaanxi National Science Foundation of China (2020JM-579), Key Research and Development Program of Shaanxi Province, China (2020GXLH-Z-032), and the Public Welfare Technology Application Research Project of Zhejiang Province, China (LGG18E020005). STEM work is supported by funding from the National Science Foundation of China (21805184), the National Science Foundation Shanghai (18ZR1425200), and the Center for High-resolution Electron Microscopy (ChEM) at ShanghaiTech University (EM02161943).

## Author contributions

G.Z. and Y.J. conceived the project and idea. L.X. and X.X. carried out the experiment and process data. G.H. and J.H. perform the theoretical calculations. Y.Y. and B.Y. carried out the STEM. All authors participated in the formulation of the paper.

## Competing interests

The authors declare no competing interests.
