## [Peer Review File · Nature Communications]

Reviewers' comments:

Reviewer #1 (Remarks to the Author):

Nice paper of pyroelectric catalysis - this is a growing area and there is little on CO₂ reduction. There is good characterisation of the materials, its pyro-catalysis and modelling to inform the mechanism. I would like to see some controls eg was the solution thermally cycled with no nano-plates and/or were the nano-plates tested at constant temperature with time; (I see this for the RhB- but was it done for the CO₂ reduction?

Bi₂WO₆ is said to show a high pyroelectric coefficient. This needs to be backed up with reference and numbers with why it is particularly advantageous compared to other materials that have been explored.

"More importantly, the suitable energy band structure and surface properties of Bi₂WO₆ allow it for CO₂ reduction" - How do the band structure and pyroelectric charge work together for CO₂ reduction? This is not clear; for example it seems to have very little link to the final mechanism in Figure 5

"The voltage produced by pyroelectric effect can be a power source for electrochemical reaction." Unusual terminology since voltage on its own to not a power - it is the potential driving force? Again this is not linked well to the modelling work or the final mechanism,

[1]
[SEP]

For the pyroelectric current measurement, is there a polarisation direction for each nano-plate since there is no poling process - this should be made clear and can be indicated with a arrow for the polarisation direction (again would link better to Figure 5).

There thermal cycling is between 15 and 70degC. What is the exact change in polarisation of the material in this range (C/m²) ? This is important as it provides the charge for the process. What would then be the voltage generated by such charge across a particle?

A stronger link between the simple mechanism of Figure 5 and the modelling in Figure 4 (for example - some additional aspects of Fig 4 could be added to Figure 5 since the final mechanism is rather simplistic).

Experimental queries:

Was the materials suspended or does it fall to the bottom of the solution. Fo the thermal cycling it is stated that "The sample is then subjected to alternating temperature-variation cycles" this could be described more detail.

Maybe of interest to consider the following overviews by Zhang in the area.

<https://doi.org/10.1016/j.joule.2019.12.019>

<https://doi.org/10.1039/C7CS00387K>

Reviewer #2 (Remarks to the Author):

The manuscript by Xiao et al. reports the use of Bi₂WO₆ nanoplates as pyroelectrocatalyst that converts CO₂ into methanol by harvesting the energy from temperature variation below 100 C. The results showed that this system could operate with high energy efficiency after 20 cycles of temperature variation. DFT calculations were performed to better understand the catalytic mechanism of CO₂ reduction.

Even though the authors claimed that they have presented a novel method for efficient conversion of

CO₂ to methanol, there are many points deserving clarification and which cannot suggest publishing in Nature Communication journal.

1. First, the authors didn't discuss possible formation of other CO₂ reduction products in their experiments. Is methanol the only product of that reaction? What is the selectivity for CH₃OH formation?

2. The authors discuss the formation of OH radical as an important intermediate in the formation of CH₃OH. However, the discussion of the actual CO₂ reduction reaction is extremely limited. This process requires 6 electrons and therefore cannot be performed in a direct one-step reaction once the OH radicals have been generated.

3. A very important point for such a catalytic paper would be the labeling test of ¹³C-CO₂, which however is missing in this manuscript.

4. The computational part discusses the addition of H ion, while it is generally accepted to have a concerted proton-electron (H⁺ + e⁻) steps, making use of the computational hydrogen electrode approach. Currently, it is not clear how the authors took into account the reduction steps.

5. Are these gas-phase calculations or is the water solvent explicitly considered? The authors say in the text that "The CO₂ reduction starts when the hydrogen ions in the solvent interact with the CO₂ molecule." This is not reflected in Figure 4.

6. How the water formed near the methanol molecule favors its release? How eventual release of methanol at steps 4 or 5 suggested by the authors and rejected based on higher activation energies possibly explain the fact that 6 electron reductions have not yet been reached?

Overall, I do not support publication in Nature Comm.

Reviewer #3 (Remarks to the Author):

This paper describes detailed characterization of Bi₂WO₆ using XRD, SEM, TEM, HR-TEM, HAADF-STEM, Piezoelectric Force microscopy, and pyrocurrent tests. The sample is tested for CO₂ reduction using Na₂SO₃ as reducing agent and also rhodamine B decomposition tests were also performed. The characterizations would attract readers, e.g, very fine HAADF-STEM images in comparison crystal structure model (Figure 1f), however, it would not be novel point of this paper. The CO₂ reduction tests are very tentative, and support data including tests in the absence of Na₂SO₃, radical trapping tests, dye decomposition tests, and DFT calculations are totally out of focus to justify the CO₂ pyroreaction data (Figure 3a and b). The reasons are described in the comments below. In my opinion, this paper does not satisfy the criteria of reliable CO₂ photo and/or pyroreduction paper.

Comments

(1) Page 2, line number 41: "biologic" should be "biological".

(2) Page 2, line numbers 43-44: The meaning of phrase is unclear: "the use of harsh/expensive reaction conditions and/or low efficiency". What is used in such cases?

(3) Page 2, last line: "to develop" is repeated. The authors need to check their manuscript.

(4) Page 4, last sentence in the Introduction section: "far" is not used for temperature. Furthermore, the grammar needs to be checked in various parts in the Introduction section.

(5) Page 6, line number 119: It is weird to mention "microscope was performed".

(6) Page 9, line number 190: The wavelength 650 nm is confusing. In Figure 3d, intense, broad peak at 710 nm accompanies a shoulder peak at 630 nm. If one of them is due to O₂⁻ radical anion, what was the source for the other peak?

(7) Page 9, line number 194: The equations 5 and 6 are not found in main text. The description would be "equations 2 - 4. However, they are the reaction of O₂ or hydroxy reduction. The reduction potential of CO₂ is different from these, and the equations 3 and 4 are not directly related to the reduction of CO₂.

(8) Page 9-10: I cannot understand why the photocomposition tests were performed for rhodamine B. It is totally unrelated to pyroreduction of CO₂.

(9) Page 12, lines number 261-262: It is extremely questionable to conclude that the CO₂ reduction was "exothermic" and to mention "spontaneous" because the formation enthalpy of CO₂ is very negative (-394 kJ/mol).

(10) Figure 5 and page 13: What happens in the absence of Na₂SO₃ (Figure 3a)? Was O₂ formation confirmed? Furthermore, ¹³CO₂-labeled confirmation test is also essential.

Response Notes (NCOMMS-20-03842)

Reviewer #1

1. Nice paper of pyroelectric catalysis - this is a growing area and there is little on CO₂ reduction. There is good characterization of the materials, its pyro-catalysis and modelling to inform the mechanism. I would like to see some controls eg was the solution thermally cycled with no nanoplates and/or were the nanoplates tested at constant temperature with time; (I see this for the RhB- but was it done for the CO₂ reduction?)

Response: Thanks for the reviewer's positive comments. Pyroelectrocatalysis is indeed rarely reported. Based on our best knowledge, up to now there is no report of pyroelectrocatalysis for CO₂ reduction.

Based on our previous experimental results (*Nanoscale* 2016, 8, 7343; *Electrochem. Commun.* 2017, 81, 124), there is no RhB dye decomposition observed under the heating-cooling cycle excitation without the additional of pyroelectrocatalyst, indicating our experiment process is originated from the pyroelectric effect of catalysts. According to the reviewer's suggestions, these comparative experiments have been added in the revised manuscript. These results show that no methanol or other products can be detected (**Fig. R1a**) under 15-70 °C variation with the absence of Bi₂WO₆ nanoplates. CO₂ is a linear molecule, which is one of the most thermodynamically stable carbon compounds, so it's hard to break the bonding of C=O. Traditional hydrogenation of CO₂ to form CH₃OH requires high temperatures (200-250 °C) and pressures (5-10 MPa) (*Nat. Commun.* 2019, 10, 5698; *ChemSusChem* 2016, 9, 322; *Fuel* 2008, 87, 443). With the presence of Bi₂WO₆ nanoplates, no methanol or other products can be detected when the test was run for 10 h at temperatures of 15 °C, 45 °C and 70 °C, respectively, as shown in **Fig. R1b**. The reason can be found in the following equation of the pyroelectric output:

$$I_{\text{pyro}} = p \cdot A \cdot (dT/dt) \quad (1)$$

where I and p are the pyroelectric current and the pyroelectric coefficient of Bi_2WO_6 , respectively. A is the Bi_2WO_6 -coated electrode area. dT/dt is the rate of temperature fluctuation. From Eqs. (1), pyroelectric charges generate only under temperature variation. If temperature is stable, there would be no pyroelectric charges, thereby no CO_2 reduces can be detected. In our experiment, the whole pyroelectrocatalytic process using wide band gap Bi_2WO_6 as catalyst was performed in dark to avoid photocatalysis.

Fig. R1 Pyroelectrocatalytic CO_2 conversion (a) without Bi_2WO_6 (b) at different stable temperature.

The relevant discussions have been added in Pages 9 of our revised manuscript.

2. Bi_2WO_6 is said to show a high pyroelectric coefficient. This needs to be backed up with reference and numbers with why it is particularly advantageous compared to other materials that have been explored.

Response: According to the reviewer's suggestions, we have added the references' comparison and explanation about the advantageous pyroelectric performance of Bi_2WO_6 . As it is well known, the ferroelectric property of Bi_2WO_6 has been reported in many articles, it has a large spontaneous polarization ($50 \mu\text{C}\cdot\text{cm}^{-2}$) and a high Curie temperature of 950 K (*J. Phys. Chem. C* 2014, 118,

13514; *Phys. Rev. B* 2004, 70, 214112). The pyroelectric coefficient of bismuth tungstate at room temperature is $1.7 \text{ nC/cm}^2\cdot\text{K}$ (*Ferroelectrics* 2002, 266, 259), which is roughly an order of magnitude with the previously-reported ZnO (*Nano Lett.* 2012, 12, 2833), but smaller than some traditional ferroelectric materials (*Energy Environ. Sci.* 2018, 11, 2198; *Nano Lett.* 2012, 12, 6408). Meanwhile, it has been reported that the pyroelectric performance is affected by the size of nanomaterials (*J. Appl. Phys.* 2010, 108, 042009). The pyroelectric coefficient strongly increases with the decreasing radius. Size-driven enhancement of pyroelectric coupling leads to giant pyroelectric current and voltage output of ferroelectric nanoparticles under temperature fluctuation (*Adv. Mater.* 2012, 24, 5357). Considering two-dimensional layer-structured Bi_2WO_6 nanoplates with relatively large specific surface, Bi_2WO_6 nanoplates is chosen as a typical pyroelectric material to testify the efficient pyroelectrocatalytic CO_2 reduction.

3. “More importantly, the suitable energy band structure and surface properties of Bi_2WO_6 allow it for CO_2 reduction “ - How do the band structure and pyroelectric charge work together for CO_2 reduction? This is not clear; for example, it seems to have very little link to the final mechanism in Fig. 5.

Response: To clarify the process of pyroelectrocatalysis, the positive and negative charges are generated from catalyst by temperature variation, which participate in the similar electrochemical reactions of CO_2 reduction to the photocatalysis (*Chem. Rev.* 2019, 119, 6, 3962; *Nat. Commun.* 2020, 11, 1443). Here, the difference between pyroelectrocatalysis and photocatalysis is the excitation source. The pyroelectric charges are generated by temperature variation of pyroelectrics. When the pyroelectrics temperature remains a stable value, pyroelectrics are initially in equilibrium with bound surface charges due to the spontaneous polarization; Once heated, the polarisation of the pyroelectric is reduced so that some of the bound charge is free to take part in reduction-oxidation reactions; When the material is subsequently cooled, the polarisation of the

material increases and charges move to the surface to balance the uncompensated screening charge carriers, leading to further reduction-oxidation reactions. During the pyroelectrocatalytic CO₂ reduction process, to satisfy the thermodynamic criteria, the reduction potential of electrons needs to be low enough. Different reduction products require different minimum potentials. For example, the electrons in conduction band should be lower than -0.38 V (vs. NHE) to generate CH₃OH in CO₂ reduction process. Many previous articles have similarly reported photocatalytic CO₂ reduction into CH₃OH using Bi₂WO₆ as catalyst (*Angew. Chem. Int. Ed.* 2015, 54, 13971; *J. Phys. Chem. C* 2016, 120, 18191; *Appl. Catal. B-Environ.* 2018, 238, 119). All address that the conduction band position of Bi₂WO₆ is suitable for CO₂ reduction to CH₃OH. Bi₂WO₆ {001} surface is a particular reactive surface energetically favoring the reduction of CO₂. Namely, CO₂ can be easily dissociated on the Bi₂WO₆ {001} surface (*ACS Appl. Mater. Interfaces* 2011, 3, 3594). More detail information can be seen on Pages 11 and 12 of theoretical calculation section in the manuscript.

4. “The voltage produced by pyroelectric effect can be a power source for electrochemical reaction. “Unusual terminology since voltage on its own to not a power - it is the potential driving force? Again this is not linked well to the modelling work or the final mechanism,

Response: According to the reviewer’s comments, we revised the link between the modelling work and the final mechanism. In addition, we corrected the terminology of power in the revised manuscript, that the voltage produced by pyroelectric effect can be a driving force for electrochemical reaction.

Pyroelectric materials' surface can generate voltage during the heating and cooling process. The pyro-potential U built in a pyroelectrocatalytic particle can be expressed as following equation,

$$U = (p \cdot \Delta T \cdot l) / \epsilon \quad (2)$$

where l , p , ΔT and ε are the size, the pyroelectric coefficient, the temperature changes and the permittivity of a pyroelectric particle, respectively (*Phys. Rev. B* 2016, 93, 195428). The pyro-potential distribution across a 2D Bi₂WO₆ nanoplate fitted by COMSOL finite element simulation is shown in **Fig. R2**, in which different colors represent different potentials. It can be seen potential difference occurs on the surfaces of the Bi₂WO₆ nanoplate.

Fig. R2 The COMSOL finite element simulation of pyro-potential of a single Bi₂WO₆ nanoplate.

Usually, the voltage output along the polarization direction can be vastly larger than the band gap of the pyroelectric material (*Phys. Chem. Chem. Phys.* 2014, 16,10408), which can be a driving force for electrochemical reactions (*Adv. Energy Mater.* 2019, 1902714). Ultimately, this driving force comes from pyroelectric effect through heating and cooling process. The reaction processes are shown in **Fig. R3**. Firstly, three H ions prefer to bind to the C atom, which breaks a C-O bond and results in a CH₃O* radical and a separate O ion (*step* “3”). Then the subsequent H ions will be attracted by the separated O ion until a H₂O molecule forms (*step* “5”). Finally, a methanol (CH₃OH) molecule is produced after one more H ion attaches to the CH₃O* radical (*step* “6”), the total equation can be expressed as following equation,

Fig. R3 Reaction processes for CO₂ reduction to CH₃OH.

5. For the pyroelectric current measurement, is there a polarization direction for each nanoplate since there is no poling process - this should be made clear and can be indicated with an arrow for the polarization direction (again would link better to Fig. 5).

Response: This is a good point. According to the previous report, the spontaneous polarization of Bi₂WO₆ is about 50 $\mu\text{C}\cdot\text{cm}^{-2}$ (*J. Phys. Chem. C* 2014, 118, 13514), and we have added arrows in Fig. 5 of our revised manuscript to clearly indicate the direction of spontaneous polarization.

All ferroelectrics are pyroelectric, where the non-centrosymmetry plays a vital role in the piezo-/pyroelectric effect. For bulk ferroelectric polycrystalline ceramics, poling is necessary to align the randomly oriented domains in order to obtain a macroscopic piezo-/pyroelectric effect. While these chemically-synthesized piezo-/pyroelectric micro/nanomaterial particles often grow along a certain crystallographic direction and can possess the crystal anisotropy and spontaneous ferroelectric polarization (*Nat. Commun.* 2010, 1, 93; *Appl. Phys. Lett.* 2006, 89, 263119; *Chem. Soc. Rev.* 2017, 46, 7757). Therefore, those nanoparticles usually behave the pyro-/piezoelectric effect without a pre-poling process (*Appl. Phys. Lett.* 2016, 109, 032904; *J. Am. Ceram. Soc.* 2011, 94, 3812; *Chem. Commun.* 2013, 49, 4003). Yu *et al.* reported that the piezoelectric constant d_{33} of BaTiO₃-based nanomaterials could be up to 13 pC/N (*Appl. Phys. Lett.* 2016, 109, 032904). Wang *et al.* reported that the d_{33} of BaTiO₃ nanowires could reach 45 pC/N (*Nano. Letts.* 2007, 7, 2966).

In addition, it has been reported that some grapheme-like two dimensional monolayer materials, such as MoS₂ can also exhibit piezoelectric effect due to the broken centro-symmetry (*Nature* 2014, 514, 470).

6. There thermal cycling is between 15 and 70 degC. What is the exact change in polarisation of the material in this range (C/m²)? This is important as it provides the charge for the process. What would then be the voltage generated by such charge across a particle?

Response: The pyroelectric coefficient of Bi₂WO₆ is 1.7 nC/cm²·K (*Ferroelectrics* 2002, 266, 259), accordingly, the change in polarization of Bi₂WO₆ between 15 °C and 70 °C is about 93.5 nC/cm². When the temperature changes, the polarization intensities of pyroelectric materials will change, and the balance between polarization and shielding charge is disturbed, so the free charges will be generated. Since Bi₂WO₆ is a wide bandgap semiconductor ($E_g = 2.69$ eV) (*ACS Appl. Mater. Interfaces* 2011, 3, 3594), which can be regarded as a capacitor. According to the following formula,

$$C = Q/U \quad (4)$$

where C , Q and U are capacitance, charge and voltage, respectively. The pyroelectric-induced charges on the Bi₂WO₆ surface can generate an electric field, resulting in a potential difference along the polarization direction.

7. A stronger link between the simple mechanism of Fig. 5 and the modelling in Fig. 4 (for example - some additional aspects of Fig. 4 could be added to Fig. 5 since the final mechanism is rather simplistic).

Response: In view of the reviewer's opinion, we have enriched the catalytic mechanism in Fig. 5 of our revised manuscript to better link the mechanism and the modelling. In Fig. 4, we calculate every reaction energy of CO₂ on the Bi₂WO₆ surface to demonstrate the possibility of CO₂ reduced

to CH₃OH. In addition, we have redrawn Fig.5 to more clearly present the pyroelectric process in the CO₂ reduction process.

8. Experimental queries: Was the materials suspended or does it fall to the bottom of the solution. For the thermal cycling it is stated that "The sample is then subjected to alternating temperature-variation cycles", this could be described more detail.

Response: In view of the reviewer's comments, we have made corresponding changes to the relevant content: High purity CO₂ gas was bubbled into the borosilicate tube for 10 min. Then the tube was immediately sealed with a rubber stopper. The sample was suspended in the solution under stirring, being applied alternating temperature between 15 °C and 70 °C in water bath. The entire catalytic process is performed in dark.

The relevant discussions have been added in Pages 17 of our revised manuscript.

Reviewer #2

The manuscript by Xiao *et al.* reports the use of Bi₂WO₆ nanoplates as pyroelectrocatalyst that converts CO₂ into methanol by harvesting the energy from temperature variation below 100 °C. The results showed that this system could operate with high energy efficiency after 20 cycles of temperature variation. DFT calculations were performed to better understand the catalytic mechanism of CO₂ reduction. Even though the authors claimed that they have presented a novel method for efficient conversion of CO₂ to methanol, there are many points deserving clarification and which cannot suggest publishing in Nature Communication journal.

Response: Thank the reviewer for the positive and constructive comments. As the reviewer said, our pyroelectrocatalyst converts CO₂ into methanol by harvesting the energy from temperature variation below 100 °C. The results showed that this system could operate with high energy efficiency after 20 cycles of temperature variation. Density functional theory (DFT) calculations were performed to better understand the catalytic mechanism of CO₂ reduction. According to the reviewer's suggestion, we have made specifically point-to-point revisions and clarifications to meet the reviewer's requirements.

1. First, the authors didn't discuss possible formation of other CO₂ reduction products in their experiments. Is methanol the only product of that reaction? What is the selectivity for CH₃OH formation?

Response: According to the reviewer's suggestion, we have conducted a comprehensive examination of the reduced products after 20 thermal cycles. For the detection of reaction products in solution, 800 μL reaction solution, 100 μL D₂O and 10 μL DMSO (0.1% vol aqueous solution) were taken into nuclear magnetic tubes, and detected with nuclear magnetic resonance (NMR) spectrometer with superconducting magnet (AVANCE NEO 400MHz, Switzerland). Here the

added DMSO is used as the internal standard reference. From the ^1H NMR spectra shown in **Fig. R1**, only methanol can be detected in liquid phase.

Fig. R1 ^1H NMR spectra of the pyroelectrocatalytic reaction solution.

The gaseous products are also analyzed by a gas chromatograph (7890B, USA) through manual injections using a thermal conductivity detector with N_2 as the carrier gas. There is only a small amount of CH_4 and CO in the gaseous product, as seen in **Fig. R2** ($0.11 \mu\text{mol}\cdot\text{g}^{-1}$ and $0.20 \mu\text{mol}\cdot\text{g}^{-1}$, respectively), indicating high selectivity of Bi_2WO_6 pyroelectrocatalytic CO_2 reduction to CH_3OH .

Fig. R2 Gas products of the pyroelectrocatalysis.

The relevant discussions have been added in Pages 8 of our revised manuscript.

2. The authors discuss the formation of OH radical as an important intermediate in the formation of CH₃OH. However, the discussion of the actual CO₂ reduction reaction is extremely limited. This process requires 6 electrons and therefore cannot be performed in a direct one-step reaction once the OH radicals have been generated.

Response: As pointed out by the reviewer, the CO₂ reduction reaction process requires 6 electrons. The reaction processes are shown in **Fig. R3**. Firstly, three H ions prefer to bind to the C atom, which breaks a C-O bond and results in a CH₃O* radical and a separate O ion (*step "3"*). Then the subsequent H ions will be attracted by the separated O ion until a H₂O molecule forms (*step "5"*). Finally, a methanol (CH₃OH) molecule is produced after one more H ion attaches to the CH₃O* radical (*step "6"*), the total equation can be expressed as following equation,

Fig. R3 Reaction processes for CO₂ reduction to CH₃OH.

In fact, the $\cdot\text{OH}$ radicals are not the intermediate in the formation of CH₃OH. The detection of $\cdot\text{OH}$ radicals illustrates the redox reactivity of positive and negative electric charges generated under temperature variation. In our process, pyroelectric induced positive and negative electric charges can react with O₂ and OH⁻, with the corresponding redox potentials -0.33 V (vs. NHE) and 2.38 V (vs. NHE) (*ACS Appl. Mater. Interfaces* 2017, 9, 2899), respectively. The reaction product O₂⁻ and $\cdot\text{OH}$ are the main active groups of catalytic degradation of dyes (*Sci. Rep.* 2020, 10, 1). The maximum reduction potential of CH₃OH is -0.38 V (vs. NHE), which is proximate with the maximum reduction potential of OH⁻ (-0.33 V vs. NHE), indicating the pyroelectrocatalytic activity for CO₂ reduction.

3. A very important point for such a catalytic paper would be the labeling test of ¹³CO₂, which however is missing in this manuscript.

Response: According to the reviewer's suggestion, we have added the isotopic labeling experiment using ¹³CO₂ as feedstock to prove the methanol is coming from CO₂ reduction. The ¹H NMR spectrum of the reaction solution in **Fig. R4a** clearly shows the formation of CH₃OH ($\delta = 3.34$ ppm) when the unlabeled CO₂ is used as feedstock. When using ¹³CO₂ instead of CO₂, the ¹H NMR spectrum of the reaction solution is shown in **Fig. R4b**. The double peaks between 3.7 and 3

ppm can be attributed to the proton coupled with the ^{13}C in $^{13}\text{CH}_3\text{OH}$ (*J. Am. Chem. Soc.* 2013, 135, 4596). The small peak of CH_3OH at 3.34 ppm possibly comes from the decomposed CO_2 of NaHCO_3 during the heating process, participating in CO_2 reduction (*J. Am. Chem. Soc.* 2014, 136, 1734). The results testify that CO_2 is the carbon source for the pyroelectrocatalytic CO_2 reduction into CH_3OH .

Fig. R4 ^1H NMR spectra of the pyroelectrocatalytic reaction solution (a) with the unlabeled CO_2 (b) with the labeled $^{13}\text{CO}_2$.

The relevant discussions have been added in Pages 10 and 11 of our revised manuscript.

4. The computational part discusses the addition of H ion, while it is generally accepted to have a concerted proton-electron ($\text{H}^+ + \text{e}^-$) steps, making use of the computational hydrogen electrode approach. Currently, it is not clear how the authors took into account the reduction steps.

Response: Intuitively, the hydrogen of the reaction in solvent exists as H^+ . However, in quantum mechanics and DFT, the electron is not a point charge but spreads in a certain region as electron cloud which is determined by its wavefunction. In DFT calculations, we deal with the wavefunctions of electrons in a specific potential originated from the nuclear charges of atoms. Therefore, it is impossible to separate hydrogen atom as proton and electron directly in DFT calculation. To model the reaction between H and the radical on the Bi_2WO_6 surface, we place a H atom beside a certain site of the radical and carried out DFT calculations to optimize the

interaction between them. Electron charge transfer happens between the H atom and radical, usually from H to the radical so that the H atom finally becomes H^+ , according to the chemical bonding between H and the radical (e.g. CO_2 molecule in this work). In other words, charge separation can be reached after self-consistent-field iterations.

The relevant discussions have been added in Pages 12 of our revised manuscript.

5. Are these gas-phase calculations or is the water solvent explicitly considered? The authors say in the text that "The CO_2 reduction starts when the hydrogen ions in the solvent interact with the CO_2 molecule." This is not reflected in Fig. 4.

Response: The calculations are based on gas phase. The solvent here carries hydrogen ions but does not involve in the reaction of CO_2 reduction. Therefore, it is not necessary to consider the water solvent in DFT calculations. In addition, we usually investigate the reaction between an individual H atom and an individual radical in DFT calculations, so it will make the calculations much more complex if we include the water solvent, and this is computationally unaffordable. In Fig. 4b, we only present the final structural configuration of each step, i.e. the process of "interaction" has been done in DFT calculation through optimizing the C-H or O-H bond length.

The relevant discussions and more explanation have been added in Pages 12 and 13 of our revised manuscript.

6. How the water formed near the methanol molecule favors its release? How eventual release of methanol at steps 4 or 5 suggested by the authors and rejected based on higher activation energies possibly explain the fact that 6 electron reductions have not yet been reached?

Response: Water molecule can form at *step "5"* but is not released from the Bi_2WO_6 surface because its release requires an activation energy of about 1.7 eV. Meanwhile, the formation of methanol at *step "6"* requires only about 0.2 eV. Therefore, the formation of methanol is much favorable than the release of water at *step "6"*. Nevertheless, the activation energies for the water

and methanol release at *step* “7” are comparable. More explicitly, the activation energy for water release is about 0.15 eV lower than that of methanol release (1.6 eV). We have added more discussion about this in Pages 12 of the revised manuscript.

At each step of calculations, we considered all possible products, where the H atom binds to the radical at different sites, and then compared the corresponding reaction energies. In Fig. 4b, we only plotted the atomic structure of the lowest-energy product or most possible product at each step. At *step* “4” or “5”, there is possible product of methanol if the H atom binds to the CH_3O^* radical rather than the O^* or OH^* radical. However, the corresponding energy is much higher than the main product. Accordingly, the production rates of methanol at *step* “4” or “5” is ignorable.

Reviewer #3

This paper describes detailed characterization of Bi_2WO_6 using XRD, SEM, TEM, HR-TEM, HAADF-STEM, Piezoelectric Force microscopy, and pyrocurrent tests. The sample is tested for CO_2 reduction using Na_2SO_3 as reducing agent and also rhodamine B decomposition tests were also performed. The characterizations would attract readers, e.g, very fine HAADF-STEM images in comparison crystal structure model (Fig. 1f), however, it would not be novel point of this paper. The CO_2 reduction tests are very tentative, and support data including tests in the absence of Na_2SO_3 , radical trapping tests, dye decomposition tests, and DFT calculations are totally out of focus to justify the CO_2 pyroreaction data (Fig. 3a and b). The reasons are described in the comments below. In my opinion, this paper does not satisfy the criteria of reliable CO_2 photo and/or pyroreduction paper.

Response: Thank the reviewer for the positive and constructive suggestions. As the reviewer's statement, we present a detailed characterization of Bi_2WO_6 using XRD, SEM, TEM, HR-TEM, HAADF-STEM, piezoelectric force microscopy, and pyrocurrent tests. The characterizations would attract readers, e.g. very fine HAADF-STEM images in comparison crystal structure model. In addition, we have carried out the direct evidence for justifying the pyroelectrocatalytic CO_2 reduction to methanol. For example, experiments with or without Bi_2WO_6 nanoplates at different temperature at 15 °C, 45 °C and 70 °C have been performed (**Fig.R1**), in which only methanol can be produced with the presence of both Bi_2WO_6 nanoplates and temperature variation. In addition, ^{13}C labeled test further proved that the carbon in methanol came from CO_2 . All of these experiments confirm the fact of pyroelectrocatalytic CO_2 reduction to methanol. We believe that our revised manuscript satisfies the criteria of reliable pyroelectrocatalytic CO_2 reduction paper.

Fig. R1 (a) pyroelectrocatalytic CO₂ conversion without Bi₂WO₆ (b) pyroelectrocatalytic CO₂ conversion at different stable temperature.

1. Page 2, line number 41: "biologic" should be "biological".

Response: The relevant content has been modified in the revised manuscript.

2. Page 2, line numbers 43–44: The meaning of phrase is unclear: "the use of harsh/expensive reaction conditions and/or low efficiency". What is used in such cases?

Response: To make the meaning of phrase clear, we have rewritten the two lines as followings: hydrogenation of CO₂ to form CH₃OH process requires high operating temperatures (200–250 °C) and high pressures (5–10 MPa), which limits the yield of methanol (*ChemSusChem* 2016, 9, 322; *Fuel* 2008, 87, 443). Photocatalytic reduction of CO₂ can be carried out in relatively mild temperature and pressure, but it always suffers drawbacks, such as insufficient light absorption and no response under dark condition (*Adv. Mater.* 2014, 26, 4607; *Appl. Catal. B-Environ.* 2019, 258, 117957). Therefore, it is imperative to develop alternative, cost-effective, and environmentally friendly approach for CO₂ conversion.

The relevant discussions have been added in Pages 2 of our revised manuscript.

3. Page 2, last line: "to develop" is repeated. The authors need to check their manuscript.

Response: The sentence has been revised in the revised manuscript.

4. Page 4, last sentence in the Introduction section: "far" is not used for temperature. Furthermore, the grammar needs to be checked in various parts in the Introduction section.

Response: The relevant content has been carefully modified in the revised manuscript.

5. Page 6, line number 119: It is weird to mention "microscope was performed".

Response: The description has been modified in the revised manuscript.

6. Page 9, line number 190: The wavelength 650 nm is confusing. In Fig. 3d, intense, broad peak at 710 nm accompanies a shoulder peak at 630 nm. If one of them is due to O₂⁻ radical anion, what was the source for the other peak?

Response: In this work, O₂⁻ was detected by Nitro Blue Tetrazolium (NBT) inspection agent, which is a widely used chemical compound for the determination of superoxide anion radical (O₂⁻) and superoxide dismutase activity in many fundamental biological processes. Its reduction reaction with superoxide and the associated spectral absorption data of monoformazan (MF) and diformazan (DF) (*Ind. Eng. Chem. Res.* 2009, 48, 9331). The reaction process can be described as following equations,

The absorption peak of MF and DF are near 630 nm and 720 nm, respectively.

The relevant discussions have been added in Pages 9 and 10 of our revised manuscript.

7. Page 9, line number 194: The equations 5 and 6 are not found in main text. The description would be "equations 2-4. However, they are the reaction of O₂ or hydroxy reduction. The

reduction potential of CO₂ is different from these, and the equations 3 and 4 are not directly related to the reduction of CO₂.

Response: On basis of the review's suggestion, we have revised the equation numbers and made a detail clarification on these equations.

In our process, pyroelectric induced positive and negative electric charges can react with O₂ and OH⁻, with the corresponding redox potentials -0.33 V (vs. NHE) and 2.38 V (vs. NHE) (*ACS Appl. Mater. Interfaces* 2017, 9, 2899), respectively. The reaction product O₂⁻ and ·OH are the main active groups of catalytic degradation of dyes (*Sci. Rep.* 2020, 10, 1). The maximum reduction potential of CH₃OH is -0.38 V (vs. NHE), which is proximate with the maximum reduction potential of OH⁻ (-0.33 V vs. NHE). Our pyroelectrocatalytic material has been demonstrated the ability for CO₂ reduction.

The relevant discussions have been added in Pages 9 of our revised manuscript.

8. Page 9-10: I cannot understand why the photocomposition (Noted: it should be pyrocomposition) tests were performed for rhodamine B. It is totally unrelated to pyroreduction of CO₂.

Response: The misunderstanding might confuse the reviewer. Except for the pyroelectrocatalysis of CO₂ reduction, we further performed RhB pyroelectrocatalytic decomposition experiment to fully demonstrate the pyroelectrocatalytic activities. In fact, RhB pyroelectrocatalytic decomposition is a visual evidence to prove the redox ability of pyroelectric generated charges.

In order to avoid confusion of CO₂ reduction and RhB decomposition, we have moved the related contents into the supporting information.

9. Page 12, lines number 261-262: It is extremely questionable to conclude that the CO₂ reduction was "exothermic" and to mention "spontaneous" because the formation enthalpy of CO₂ is very negative (-394 kJ/mol).

Response: As the reviewer said, the formation enthalpy of CO₂ is very negative. The binding energy on the Bi₂WO₆ surface is -3.6 eV per molecule in our calculation. Nevertheless, the interaction between H and CO₂ is still very strong. As can be seen in **Fig. R2**, the first four reaction steps are exothermic with large negative reaction energies. Even through the last three reaction steps are endothermic, the overall process of the CO₂ reduction is still exothermic as shown in **Fig. R2**. Therefore, the activation energies required in the last three reaction steps can be compensated by the energy released in the first four steps. Therefore, the CO₂ reduction lasts spontaneously in principle. However, given the energy loss in a solvent environment, energy supply is still needed but the energy demanding is not much. This is reason that the temperature is not high in our experiments.

We have rewritten these sentences on Page 13 to explain this point.

Fig. R2 Reaction energies for the CO₂ reduction. Eight reaction steps are considered. The reaction energies for the side product (methanol) at steps “4” and “5” are indicated by short dashed lines.

The short red (dashed) lines are estimated from the free energies.

10. Fig. 5 and page 13: What happens in the absence of Na₂SO₃ (Fig. 3a)? Was O₂ formation confirmed? Furthermore, ¹³CO₂-labeled confirmation test is also essential.

Response: To solve the problems of carriers' recombination, sacrificial electron donors have been frequently added into the reaction system to consume the generated holes, thereby increasing the survival time of generated electrons. It includes two steps: Firstly, electric charge carriers are generated under light or thermal signals *via* photovoltaic or pyroelectric effect, respectively. Secondly, the generated positive and negative electric carriers induce the chemical oxidation-reduction reactions. The position of the valence band of Bi₂WO₆ conditions satisfy for oxygen production (*Appl. Catal. B-Environ.* 2017, 219, 209; *Appl. Catal. B-Environ.* 2018, 238, 119). It has been similarly reported that CO₂ reduction using Bi₂WO₆ as catalyst generated oxygen when there is no sacrificial agent added. Kakekhani *et al.* have theoretically proved that the pyroelectrocatalytic product of water splitting includes not only H₂, but also O₂ (*J. Mater. Chem. A* 2016, 4, 5235). Therefore, oxygen will be produced in the pyroelectrocatalytic process without the addition of sacrificial agents.

According to the reviewer's suggestion, we added the isotopic labeling experiment using ¹³CO₂ as feedstock in order to prove that CH₃OH is the product of CO₂ reduction. The ¹H NMR spectrum of the reaction solution in **Fig. 3a** clearly showed the formation of CH₃OH ($\delta = 3.34$ ppm) when the unlabeled CO₂ was used as feedstock. **Fig. 3b** shows the ¹H NMR spectrum of the reaction solution when using ¹³CO₂ instead of CO₂. The double peaks between 3.7 and 3 ppm can be attributed to the proton coupled with the ¹³C in ¹³CH₃OH (*J. Am. Chem. Soc.* 2013, 135, 4596). The small peak of CH₃OH at 3.34 ppm possibly comes from the decomposed CO₂ of NaHCO₃ in the reaction solution, participating in CO₂ reduction during the heating process (*J. Am. Chem. Soc.* 2014, 136, 1734). The results testify that CO₂ is the carbon source for the pyroelectrocatalytic CO₂ reduction into CH₃OH.

Fig. R3 ^1H NMR spectra of the pyroelectrocatalytic reaction solution (a) with the unlabeled CO_2
(b) with the labeled $^{13}\text{CO}_2$.

The relevant discussions have been added in Pages 10 and 11 of our revised manuscript.

(End)

REVIEWER COMMENTS

Reviewer #1 (Remarks to the Author):

I am happy with the corrections in response to reviewer comments and the paper is improved.

Reviewer #2 (Remarks to the Author):

The authors have significantly improved their manuscript taking into account all questions and concerns raised by the referees. I suggest publication after polishing some repetitions and improving the English (for instance, page 13 lines 269-273 and page 14 lines 291-294).

Reviewer #3 (Remarks to the Author):

This paper reports catalysis when the temperature change cycle of catalyst was repeated. The data is only one repetition condition, and is not enough to prove pyro-originated catalysis. Directly related characterization is lacking.

Comments

(1) Page 3, lines 1-2: It is not fair comparison. Photocatalysts can be fully irradiated under light. It is not always rainy days on the earth.

(2) Page 3, line 3 from the bottom: The language is not improved. It should be "The study has been reported for seventy years." Page 7, line 10: "can generation" would be "can generate".

(3) The catalytic result section: Rather than to demonstrate both CO₂ and Rhodamine B decompositions, it is essential to prove the temperature derivation (the rate and frequency, the variation of temperature range) was the origin of the catalysis.

(4) Figure 3: Why was a half of ¹²CH₃OH observed in the ¹³CO₂-labeled test? Just one point chart in the reaction test is insufficient.

(5) The extent of pyro-derived change separation, e.g. in Figure 5, is not experimentally demonstrated in this paper.

Response letter

Dear Dr. Adam Weingarten and Referees,

We submit our 2nd revised manuscript (Based on NCOMMS-20-03842A-Z) entitled “Pyroelectrocatalytic CO₂ reduction for methanol driven by temperature-variation” to *Nature Communications*.

We express our gratitude to you and the three reviewers for their valuable comments on our original and 1st revised manuscript. We have carefully revised our manuscript based on the reviewers’ suggestions and the main changes are marked in the revised manuscript. The point to point response is attached as followings:

Reviewer #1

I am happy with the corrections in response to reviewer comments and the paper is improved.

Response: Thanks.

Reviewer #2

The authors have significantly improved their manuscript taking into account all questions and concerns raised by the referees. I suggest publication after polishing some repetitions and improving the English (for instance, page 13 lines 269-273 and page 14 lines 291-294)

Response: Thanks for the reviewer's comments. We have improved the English in the revised manuscript.

Reviewer #3

This paper reports catalysis when the temperature change cycle of catalyst was repeated. The data is only one repetition condition, and is not enough to prove pyro-originated catalysis. Directly related characterization is lacking.

Response: According to the reviewer's comment, we have strengthened the experimental proof (temperature derivation including the rate and the variation of temperature range) on the pyro-originated catalysis in the revised manuscript. The details are described in the following response (3). In addition, we have added and provided direct characterizations to support the process of pyroelectrocatalysis in the revised manuscript. Piezoelectric force microscope characterization indicates the ferroelectricity of Bi_2WO_6 (Fig. 2a-d), Pyroelectric current measurement (Fig. 2e-f), free radical detections (Fig. 3c-d) and ESR characterization (Fig. S8) prove the generation and separation of pyroelectric charges. Isotope labeling experiments further demonstrate the occurrence of pyroelectrocatalytic CO_2 reduction to methanol (Fig. 3e-f). Comparative experiments on pyroelectrocatalysis in Fig.3a-b and Fig. S7 shows the CO_2 reduction reaction can only occur when both the temperature variation and the catalyst are present at the same time. Density functional theory calculations have also been performed to better understand the pyroelectrocatalytic mechanism of CO_2 reduction (Fig. 4). All these experimental results and theoretical calculations have confirmed the pyro-originated pyroelectrocatalytic CO_2 reduction of Bi_2WO_6 under temperature variation.

(1) Page 3, lines 1-2: It is not fair comparison. Photocatalysts can be fully irradiated under light. It is not always rainy days on the earth.

Response: Photocatalysis, as an advanced method, can be carried out in mild temperature and pressure and have attracted much attention due to its potential application in converting CO_2 into hydrocarbon fuel. The use of clean and renewable solar energy changing CO_2 into hydrocarbon fuel by photocatalysis can not only achieve the purpose of energy saving and suppression of the greenhouse effect, but also alleviate the problem of energy shortage. As a matter of fact, photocatalytic reduction of CO_2 can be carried out in mild temperature and pressure, but it does not work in dark (*Chem. Soc. Rev.* 2020, 49, 2937; *Energy Environ. Sci.* 2018, 11, 2198). As for

pyroelectrocatalysis, temperature variation in daily life is used for clean energy in our work. It would be desirable to utilize the natural temperature variation for CO₂ conversion. This article introduces a new method of CO₂ reduction through temperature variation even in dark.

Taking the reviewer's suggestion into account, the relevant content in Page 3, lines 1-2 of the manuscript has been rewritten to make the readers easy to understand the difference between photocatalysis and pyroelectrocatalysis in utilizing the environmental clean solar light energy and the cold-hot alternation thermal energy.

(2) Page 3, line 3 from the bottom: The language is not improved. It should be "The study has been reported for seventy years." Page 7, line 10: "can generation" would be "can generate".

Response: Thanks for the reviewer's comments. We have carefully edited the whole manuscript to enhance English.

(3) The catalytic result section: Rather than to demonstrate both CO₂ and Rhodamine B decompositions, it is essential to prove the temperature derivation (the rate and frequency, the variation of temperature range) was the origin of the catalysis.

Response: Thanks for the reviewer's comments. Temperature derivation (the rate and frequency, the variation of temperature range) is very important influence factors on pyroelectrocatalysis.

In general, a high rate of temperature change ($\partial T/\partial t$) is beneficial for pyroelectricity. It has been reported that the pyroelectric charge is proportional to $\partial T/\partial t$ in the range of 10⁻² to 10⁶ °C/s (*J. Appl. Phys* 2012, 112, 104106). Here, we have investigated the effect of temperature change rate on the pyroelectrocatalytic degradation dyes in **Fig. R1**, indicating the more rapid speed of degradation along with the higher values of $\partial T/\partial t$.

Fig. R1 Pyroelectrocatalytic dye degradation under different temperature change rates.

The range of temperature can also affect the pyroelectrocatalytic CO₂ reduction. **Fig. R2** shows the yield of methanol after 10 thermal cycles in different temperature ranges (15-40 °C, 15-50 °C, 15-70 °C, 15-85 °C). In **Fig. R2**, the methanol yield increases as the temperature increases. The pyro-induced charges (dQ) can be expressed in Eq. (1)

$$dQ = p \cdot A \cdot dT \quad (1)$$

where p is the pyroelectric coefficient of Bi₂WO₆. A is the Bi₂WO₆-coated electrode area. dT is the temperature range. The large range of temperature can produce more pyrocharges, resulting in the better pyroelectrocatalytic performance. The relevant discussions have been added to the revised manuscript.

Fig. R2 Methanol yield of different temperature ranges.

(4) Figure 3: Why was a half of $^{12}\text{CH}_3\text{OH}$ observed in the $^{13}\text{CO}_2$ -labeled test? Just one point chart in the reaction test is insufficient.

Response: In the catalytic CO_2 reduction systems, NaHCO_3 or KHCO_3 is often used as a buffer (*J. Am. Chem. Soc.* 2017, 139, 16161; *J. Am. Chem. Soc.* 2014, 136, 1734; *Adv. Mater.* 2019, 31, 1804710; *Adv. Mater.* 2019, 31, 1804710). The $^{12}\text{CO}_2$ is generated from the heating of HCO_3^- ions. Therefore, both $^{13}\text{CO}_2$ and $^{12}\text{CO}_2$ participate in the catalytic reactions to form methanol in the ^{13}C isotope experiment, which was also observed in the $^{13}\text{CO}_2$ -labeled test (*Joule* 2018, 2, 1369). To avoid $^{12}\text{CO}_2$, we have completed the additional experiment using NaOH instead of NaHCO_3 . **Fig. R3** shows the NMR spectra of the pyroelectrocatalytic reaction solution with labeled $^{13}\text{CO}_2$ using NaOH as buffer. There is only $^{13}\text{CH}_3\text{OH}$ signal in **Fig. R3**, confirming that the methanol originates from the pyroelectrocatalytic CO_2 reduction. In addition, the more points' chart has been presented in the Fig.S3 of supporting information. The relevant discussions have been added to the revised manuscript.

Fig. R3 ^1H NMR spectra of the pyroelectrocatalytic reaction solution with labeled $^{13}\text{CO}_2$.

(5) The extent of pyro-derived charge separation, e.g. in Figure 5, is not experimentally demonstrated in this paper.

Response: We have presented the experiments to demonstrate the pyro-derived charge separation (**Fig. R4 and 5**). **Fig. R4** shows that $\cdot\text{OH}$ and O_2^- are detected by fluorescence spectrometry using

terephthalic acid as a photoluminescent $\cdot\text{OH}$ trapping agent and by UV-Vis spectrophotometer using nitro-blue tetrazolium as absorption $\text{O}_2^{\cdot-}$ trapping agent, respectively.

Fig. R4 (a) The fluorescence spectra of 2-hydroxyterephthalic acid, (b) The absorption spectra of diformazan and monoformazan.

We have further measured the electron spin resonance (ESR) signals of radical's spin-trapped by 5,5-dimethyl-1-pyrroline-N-oxide to verify the generation of the $\cdot\text{OH}$ and $\text{O}_2^{\cdot-}$. **Fig. R5** shows the characteristic peaks of DMPO- $\cdot\text{OH}$ and DMPO- $\text{O}_2^{\cdot-}$ spin adducts with the increase of thermal cycles, indicating the amount increase of both $\cdot\text{OH}$ and $\text{O}_2^{\cdot-}$. In contrast, there is no radical signals without the thermal cycles. Moreover, the pyro-current response in Fig. 2f-g can also demonstrate the pyro-derived charge separation.

Fig. R5 ESR spectra of (a) DMPO- $\cdot\text{OH}$ and (b) DMPO- $\text{O}_2^{\cdot-}$ adducts after undergoing different thermal cycles.

As it is well known, pyroelectric materials can generate the positive and negative charges under cold-hot temperature variation (*Nat. Mater.* 2018, 17, 432). The temperature change of the pyroelectric material causes the imbalance between its spontaneously polarization and surface charges, which results in the generation of free charge. The pyro-induced positive and negative charges can react with O₂ and OH⁻ in water to form O₂⁻ and ·OH, respectively, as shown in Eq. (2)-(4).

The relevant discussions have been added to our revised manuscript.

REVIEWER COMMENTS

Reviewer #3 (Remarks to the Author):

This paper reports the EM characterization of layered Bi₂WO₆ and the feasibility of pyrocatalysis for CO₂ reduction. This reviewer does not recommend this paper for publication due to following reasons.

Pages 1-7: The language is still difficult to read. Minor errors still remain, e.g. Fig. 2e instead of Fig. 2f (page 7, line 6 from the bottom) and Fig. 2f instead of Fig. 2g (page 7, line 5 from the bottom).

Pages 8-11: Product O₂ is not monitored even qualitatively. It is doubtful why methanol was always happily produced selectively. The impurity transformation needs to be carefully checked. The same opinion is repeated. One point check of isotope-label test in the time course of pyrocatalysis is insufficient because methanol is often formed starting from impurity, and time course monitoring is essential to prove the reaction

Response letter

Dear Dr. Adam Weingarten,

We are very grateful to you and the reviewer's valuable comments on our manuscript. After three reviewers' comments and third rounds' revisions, we believe that we have fully addressed the reviewers' comments/suggestions about this work. According to the comments of the reviewer #3, we have further added specific experiments including strictly controlled carbon source in the catalysis process (only use the isotope $^{13}\text{CO}_2$) and the time dependence isotope-label test. The results show the methanol is not from impurities but only from CO_2 reduction. The point to point response is attached as followings:

Reviewer #3

This paper reports the EM characterization of layered Bi_2WO_6 and the feasibility of pyrocatalysis for CO_2 reduction. This reviewer does not recommend this paper for publication due to following reasons.

(1) Pages 1-7: The language is still difficult to read. Minor errors still remain, e.g. Fig. 2e instead of Fig. 2f (page 7, line 6 from the bottom) and Fig. 2f instead of Fig. 2g (page 7, line 5 from the bottom).

Response: Thanks for reviewer's comment. The relevant content has been carefully modified and highlighted in the revised manuscript.

(2) Pages 8-11: Product O_2 is not monitored even qualitatively. It is doubtful why methanol was always happily produced selectively. The impurity transformation needs to be carefully checked. The same opinion is repeated. One-point check of isotope-label test in the time course of pyrocatalysis is insufficient because methanol is often formed starting from impurity, and time course monitoring is essential to prove the reaction

$$\text{CO}_2 + \text{H}_2\text{O} \rightarrow \text{CH}_3\text{OH} + 3/2 \text{O}_2$$

Response: According to the reviewer's suggestion, we have added O_2 monitor experiments (Figure R1) and many thermal cycles of isotope-label test in the time course of pyroelectrocatalysis (Figure R2). Our experimental results verify the CH_3OH is totally from CO_2 reduction not from any impurity. Usually, organic impurities in

carbon-based catalysts can possibly be oxidized to hydrocarbons (*Nat. Commun* 2020, 11, 1). Nevertheless, our catalyst is inorganic Bi_2WO_6 without carbon. As for the whole reaction process, we have carefully checked and carried out each step experiment and test. Oxygen as a byproduct of pyroelectrocatalytic CO_2 reduction is measured by gas chromatography (Fuli GC9790plus). Fig. R1 shows the amount of O_2 increases with the number of thermal cycles. The amount of oxygen are roughly proportional to the amount of methanol.

Fig. R1 The yield of O_2 with different thermal cycles.

The time dependence isotope-label $^{13}\text{CO}_2$ test is shown in Fig. R2. Before the reaction starts, neither $^{12}\text{CH}_3\text{OH}$ nor $^{13}\text{CH}_3\text{OH}$ are detected. Then, the yield of methanol increases as the increase of thermal cycles, moreover, the signal of $^{12}\text{CH}_3\text{OH}$ is not detected throughout the whole catalytic reaction. Here, we have strictly controlled the carbon source in the catalysis process, no other carbon source is added except $^{13}\text{CO}_2$. The above data shows that methanol is not generated from impurities but from CO_2 reduction.

Fig. R2 NMR of isotope-label test with cumulative thermal cycling over time.

There are many reports regarding methanol to be produced from CO₂ through different catalytic processes (*Nat. Commun.* 2020, 11, 2531; *Angew. Chem. Int. Ed.* 2019, 58, 3804; *Chem. Rev.* 2019, 119, 3962–4179; *Joule* 2018, 2, 1369). As it is well known, the enthalpy change (ΔH) of CO₂ to methane (890 kJ/mol) is higher than that of methanol (727 kJ/mol) (*J. Mater. Chem. A*, 2018, 6, 22411). Meanwhile, molecular absorption and desorption have a great impact on the selectivity of catalysis (*ACS Catal.* 2020, 10, 8632; *Nano Energy* 2020, 75, 104959). As for Bi₂WO₆, ferroelectric polarization can affect molecular adsorption and desorption from the surface of the materials (*J. Mater. Chem. A* 2016, 4, 5235). Moreover, it has been reported that the dissociated CO + O on the Bi₂WO₆ surface is easily recombined to CO₂ (*ACS Appl. Mater. Interfaces* 2011, 3, 3594). All these points suggest methanol selectivity in our pyroelectrocatalytic process.

Reviewers' comments:

Reviewer #3 (Remarks to the Author):

This paper reports characterization of Bi₂WO₆ for pyroelectric catalysis and the application for CO₂ conversion is also reported. The reviewer judge that the steady methanol formation from CO₂ is not proved in this paper, and do not recommend the publication in Nature Communications as commented below. At least, information is insufficient, and the content is in appropriate as a short Communication.

Comments:

(1) Writing is not on the standard level for Nature Communications. Page 2, "more release of carbon dioxide in atmosphere" is not strict. In next sentence, "energy crisis" is not "challenge" but a problem. Page 3, first line, "in mild temperature and pressure" should be "at mild temperature and pressure". page 5, line 9, "lattice plane distances" should be "lattice plane intervals", and on next line, the abbreviation of "scanning TEM" is not noted,

(2) Page 4, lines 10-11: The rate is 17 micro mol/g/h that is in lowest range for photocatalytic conversion of CO₂. Much more caution is needed for the possibility of impurity conversion rather than CO₂ for the chemical reaction on the lowest range.

(3) Page 10 and Figure 3: I am still not confident about the isotope-labelled study. As the amount of methanol is small (comment 2), the signal/background ratio is low in Figure 3e and f. Based on the quality of NMR data, it is insufficient to conclude the steady formation of ¹³CH₃OH. The time course change of the intensity is indispensable during the authors' pyroelectrocatalytic CO₂ reduction tests.

(4) Response letter, Figure R1: The experimental conditions are not noted, and this reviewer cannot compare the data to methanol formation data (Figure 3). If it is data in the presence of Na₂SO₃, the formed O₂ amount is not 1.5 times of formed methanol amount. The reason is unclear. Figure R1 cannot be found in main text nor Supplementary Information and the readers of this Journal cannot see it.

(5) Page 17, Experimental: Does 0.1 volume% DMSO not affect proton NMR (Figure 3f)? In lines 5-6, was high purity ¹³CO₂ gas really bubbled into the borosilicate tube for 10 min in the labelled test?

(6) Page 15, Experimental: The purity of 99.9% is chemical purity, but isotope (¹³C) purity needs to be added and evaluated based on the analysis (Figure 3). If it is difficult, the reaction mechanism from ¹³CO₂ to ¹³CH₃OH needs to be described much more detail for readers' interest. The conversion comprises several reaction steps and some does not proceed by the effects of charge separation. Such discussion is not included in this paper and steady methanol formation is doubtful.

REVIEWERS' COMMENTS

Reviewer #4 (Remarks to the Author):

The authors report on the production of methanol from pyroelectrocatalytic (or pyroelectric catalytic to avoid confusion with standard electrocatalysis?) reduction of CO₂. Their manuscript has gone through previous rounds of review, and the goal of this review is to establish if the problems encountered during these previous rounds of review are substantial enough to hinder publication of this manuscript.

First off, I agree with the initial comments of the reviewers that the manuscript needs English editing (authors have indicated this will be done at a later stage). A substantial concern has been the source of carbon for methanol production. The authors have done a plethora of control experiments to show that no methanol is produced without Bi₂WO₆ and at constant temperature. Also, the authors have carried out ¹³C CO₂ labeled experiments. I understand concerns in earlier rounds of review, as the results showed partial production of ¹²C and ¹³C methanol (probably due to the use (H)¹²CO₃⁻ in the electrolyte), but the results presented by the authors in Figures 3e and f, and S11 provide clear evidence to me that observed methanol is without a doubt produced from CO₂. Therefore, I do not see any fundamental scientific issues that should hold back the publication of this manuscript.

One concern that remains for me is that the description given in the Experimental section on experimental details is very brief. I understand that this is required for a communication style article, but the authors should use the SI to provide more insight into how they performed their experiments. At this stage, only figures are provided in the SI, which is in my opinion a wasted opportunity.

Dear Dr. Adam Weingarten and Reviewer,

We submit our revised manuscript entitled “*Pyroelectrocatalytic CO₂ reduction for methanol driven by temperature-variation*” (NCOMMS-20-03842D-Z) to *Nature Communications*. We appreciate for you and reviewers for their valuable comments on our original manuscript. After first round of revisions according to the reviewers’ comments, Reviewer 1 and Reviewer 2 suggest publication as follows; Reviewer #1 (Remarks to the Author): I am happy with the corrections in response to reviewer comments and the paper is improved. Reviewer #2 (Remarks to the Author): The authors have significantly improved their manuscript taking into account all questions and concerns raised by the referees. I suggest publication after polishing some repetitions and improving the English (for instance, page 13 lines 269-273 and page 14 lines 291-294). After four rounds of revisions according to the comments of reviewer 3, we have carefully carried out additional experiments and fully addressed his/her comments/suggestions. However, we found that reviewer 3 has had the personal bias to reject the manuscript from the beginning. Reviewer 3 never really wanted to improve the manuscript, and only motivation was to reject our research manuscript without compelling reason. It is unreasonable that reviewer 3 made judgement on the basis of not the solid experimental data but his/her own feelings and emotions (The specific evidence is listed in the following response). All of our authors hope to be treated fairly, objectively, scientifically.

Point-to-point responses to all comments and suggestions raised by reviewer 3 are provided following this letter. We also attach the previous three rounds of revisions for your reference.

The point-to-point responses to reviewer’s comments

Fourth round of Reviewer #3

This paper reports characterization of Bi₂WO₆ for pyroelectric catalysis and the application for CO₂ conversion is also reported. The reviewer judge that the steady methanol formation from CO₂ is not proved in this paper, and do not recommend the publication in Nature Communications as commented below. At least, information is insufficient, and the content is in appropriate as a short Communication.

Response: In fact, we have completely proved the steady methanol formation from CO₂ in our manuscript. For example, to directly demonstrate the methanol from CO₂ reduction, we have completed specific experiments and analyses including the isotope-label ¹³CO₂ comparison with ¹²CO₂ (Fig. 3e-f), the yield of O₂ with different thermal cycles without Na₂SO₃ (Fig. S5), NMR of isotope-label test with cumulative thermal cycling over time (Fig. S11), and the following experimental data (Fig. 2f-g, Fig. 3c-d and Fig. S9: generation and separation of pyroelectric charges, Fig S6 and Fig .S8: Comparative tests under different experimental conditions) and theoretical simulations (Fig. 4: pyroelectrocatalytic mechanism of CO₂ reduction). However, reviewer 3 non-scientifically commented that the steady methanol formation from CO₂ was not

proved in this paper.

Fig. 3 (e) ¹H NMR spectra of the pyroelectrocatalytic reaction solution with unlabeled CO₂ and (f) ¹H NMR spectra of the pyroelectrocatalytic reaction solution with labeled ¹³CO₂.

Fig. S5 Yield of Methanol and Oxygen without Na₂SO₃

Fig. S11 NMR of isotope-label test with cumulative thermal cycling over time

Fig.2 (f) The Pyro-current response of Bi₂WO₆, (g) enlarged view of one full light on/off cycle is shown in Fig. f.

Fig. 3 (c) The fluorescence spectra of 2-hydroxyterephthalic acid, (d) The absorption spectra of diformazan and monoformazan.

Fig. S9 Electron spin resonance spectra of (a) DMPO-·OH and (b) DMPO-O₂^{·-} adducts after undergoing different thermal cycles.

Fig. S6 Methanol yield of different temperature change ranges.

Fig. S8 Pyroelectrocatalytic CO_2 reduction (a) without Bi_2WO_6 (b) with Bi_2WO_6 at different stable temperature.

Fig. 4 Adsorption configurations and reaction path of CO_2 into CH_3OH on Bi_2WO_6 . (a) Five different adsorption configurations for CO_2 on Bi_2WO_6 . (b) Structures and (c) reaction energies for the CO_2 reduction. Eight reaction steps are considered. The red, cyan, light grey and dark grey spheres stand for O, W, Bi, H and C atoms. Only the atoms around CO_2 are highlighted. The reaction energies for the side product (methanol) at steps "4" and "5" are indicated by short dashed lines. The short red (dashed) lines are estimated from the free energies.

Comments:

(1) Writing is not on the standard level for Nature Communications. Page 2, "more release of carbon dioxide in atmosphere" is not strict. In next sentence, "energy crisis" is not "challenge" but a problem. Page 3, first line, "in mild temperature and pressure" should be "at mild temperature and pressure". page 5, line 9, "lattice plane distances" should be "lattice plane intervals", and on next line, the abbreviation of "scanning TEM" is not noted.

Response: As for the English, we will use the nature research editing service to polish our manuscript. In addition, the relevant content has been carefully modified in the revised manuscript.

(2) Page 4, lines 10-11: The rate is 17 micro mol/g/h that is in lowest range for photocatalytic conversion of CO₂. Much more caution is needed for the possibility of impurity conversion rather than CO₂ for the chemical reaction on the lowest range.

Response: In fact, organic impurity can possibly be oxidized to hydrocarbons, but impurity conversion can only happen in carbon-based catalysts (*Nat. Commun* 2020, 11, 1). However, Bi₂WO₆ is inorganic catalyst, we didn't use any organic sacrificial agents in the whole reaction process, so methanol is impossibly converted from impurities. Furthermore, the isotope detection experiment directly proved that the production of methanol comes from the reduction of CO₂. Using ¹³CO₂ as feedback, the ¹H NMR spectrum of the reaction solution in **Fig. R1b** shows doublet peaks between 3.7 and 3.0 ppm, which is attributed to the proton coupled with the ¹³C of ¹³CH₃OH (*J. Am. Chem. Soc.* 2013, 135, 4596-4599), while the ¹H NMR spectrum of the ¹²CO₂ reaction solution (**Fig. R1a**) shows the formation of ¹²CH₃OH ($\delta=3.34$ ppm).

Fig. R1 ¹H NMR spectra of the pyroelectrocatalytic reaction solution with (a) unlabeled CO₂ and (b) labeled ¹³CO₂.

(3) Page 10 and Figure 3: I am still not confident about the isotope-labelled study. As the amount of methanol is small (comment 2), the signal/background ratio is low in Figure 3e and f. Based on the quality of NMR data, it is insufficient to conclude the steady formation of ¹³CH₃OH. The time course change of the intensity is indispensable during the authors' pyroelectrocatalytic CO₂ reduction tests.

Response: Scientifically speaking, isotope labeling experiment is sufficient to prove that the production of methanol comes from CO₂ reduction in this work. The time course change of the NMR intensity is shown in **Fig. R2**, which we have shown in the third round of response letter. Further, except the isotope labeling experiment, there are other experiments to prove methanol is produced from pyroelectrocatalytic CO₂ reduction. In Fig.3a-b, the production of methanol is measured by gas chromatography. Comparative experiments in Fig. S7 shows the CO₂ reduction reaction can only occur when both the temperature variation and the catalyst are present at the same time. Free radical detections (Fig. 3c-d) and ESR characterization (Fig. S8) prove the generation and separation of pyroelectric charges. Density functional theory calculations have also been performed to better understand the pyroelectrocatalytic mechanism of CO₂ reduction (Fig. 4). All these experimental results and theoretical calculations have confirmed the pyro-originated

pyroelectrocatalytic CO₂ reduction of Bi₂WO₆ under temperature variation.

Fig. R2 NMR of isotope-label test with cumulative thermal cycling over time.

(4) Response letter, Figure R1: The experimental conditions are not noted, and this reviewer cannot compare the data to methanol formation data (Figure 3). If it is data in the presence of Na₂SO₃, the formed O₂ amount is not 1.5 times of formed methanol amount. The reason is unclear. Figure R1 cannot be found in main text nor Supplementary Information and the readers of this Journal cannot see it.

Response: Na₂SO₃ was not added in the O₂ production detection experiment, this because Na₂SO₃ can react with positive charges to reduce O₂ formation. Without Na₂SO₃, the amount of O₂ is roughly 1.5 times the amount of methanol, which can be seen in **Fig. R3**. In view of the reviewer's comments, we put it in the Supplementary Information.

Fig. R3 Yield of Methanol and Oxygen without Na₂SO₃

(5) Page 17, Experimental: Does 0.1 volume% DMSO not affect proton NMR (Figure 3f)? In lines 5-6, was high purity ¹³CO₂ gas really bubbled into the borosilicate tube for 10 min in the labelled test?

Response: In Fig. 3e and Fig. 3f, both the related experiment added 0.1 volume% DMSO as the internal standard. The doublet peaks between 3.7 and 3.0 ppm is attributed to the proton coupled with the ¹³C of ¹³CH₃OH, but not DMSO. In addition to the ¹³CO₂ in the solution, there is a large amount of ¹³CO₂ above the solution in the vessel. In order to ensure the purity of the ¹³CO₂ in the reaction vessel, we vacuum the vessel and then fill it with high purity ¹³CO₂.

(6) Page 15, Experimental: The purity of 99.9% is chemical purity, but isotope (^{13}C) purity needs to be added and evaluated based on the analysis (Figure 3). If it is difficult, the reaction mechanism from $^{13}\text{CO}_2$ to $^{13}\text{CH}_3\text{OH}$ needs to be described much more detail for readers' interest. The conversion comprises several reaction steps and some does not proceed by the effects of charge separation. Such discussion is not included in this paper and steady methanol formation is doubtful.

Response: The isotope labeling experiment is to prove that the carbon in methanol comes from CO_2 , the purity of 99 % is able to achieve this purpose (*J. Am. Chem. Soc.* 2013, 135, 4596-4599), and the purity of $^{13}\text{CO}_2$ used in our experiments is 99.9%. The reaction processes are shown in **Fig. R4** (The detailed reaction process of CO_2 to methanol has already presented in our initial manuscript Page13, Lines4-11). Firstly, three H ions prefer to bind to the C atom, which breaks a C-O bond and results in a CH_3O^* radical and a separate O ion (*step "3"*). Then the subsequent H ions will be attracted by the separated O ion until a H_2O molecule forms (*step "5"*). Finally, a methanol (CH_3OH) molecule is produced after one more H ion attaches to the CH_3O^* radical (*step "6"*)

Fig. R4 Reaction processes for CO_2 reduction to CH_3OH .

All three reviewers' comments and responses

First round of Reviewer #1

1. Nice paper of pyroelectric catalysis - this is a growing area and there is little on CO₂ reduction. There is good characterization of the materials, its pyro-catalysis and modelling to inform the mechanism. I would like to see some controls eg was the solution thermally cycled with no nanoplates and/or were the nanoplates tested at constant temperature with time; (I see this for the RhB- but was it done for the CO₂ reduction?)

Response: Thanks for the reviewer's positive comments. Pyroelectrocatalysis is indeed rarely reported. Based on our best knowledge, up to now there is no report of pyroelectrocatalysis for CO₂ reduction.

Based on our previous experimental results (*Nanoscale* 2016, 8, 7343; *Electrochem. Commun.* 2017, 81, 124), there is no RhB dye decomposition observed under the heating-cooling cycle excitation without the additional of pyroelectrocatalyst, indicating our experiment process is originated from the pyroelectric effect of catalysts. According to the reviewer's suggestions, these comparative experiments have been added in the revised manuscript. These results show that no methanol or other products can be detected (**Fig. R1a**) under 15-70 °C variation with the absence of Bi₂WO₆ nanoplates. CO₂ is a linear molecule, which is one of the most thermodynamically stable carbon compounds, so it's hard to break the bonding of C=O. Traditional hydrogenation of CO₂ to form CH₃OH requires high temperatures (200-250 °C) and pressures (5-10 MPa) (*Nat. Commun.* 2019, 10, 5698; *ChemSusChem* 2016, 9, 322; *Fuel* 2008, 87, 443). With the presence of Bi₂WO₆ nanoplates, no methanol or other products can be detected when the test was run for 10 h at temperatures of 15 °C, 45 °C and 70 °C, respectively, as shown in **Fig. R1b**. The reason can be found in the following equation of the pyroelectric output:

$$I_{\text{pyro}} = p \cdot A \cdot (dT/dt) \quad (1)$$

where I and p are the pyroelectric current and the pyroelectric coefficient of Bi₂WO₆, respectively. A is the Bi₂WO₆-coated electrode area. dT/dt is the rate of temperature fluctuation. From Eqs. (1), pyroelectric charges generate only under temperature variation. If temperature is stable, there would be no pyroelectric charges, thereby no CO₂ reduces can be detected. In our experiment, the whole pyroelectrocatalytic process using wide band gap Bi₂WO₆ as catalyst was performed in dark to avoid photocatalysis.

Fig. R1 Pyroelectrocatalytic CO₂ conversion (a) without Bi₂WO₆ (b) at different stable temperature.

The relevant discussions have been added in Pages 9 of our revised manuscript.

2. Bi₂WO₆ is said to show a high pyroelectric coefficient. This needs to be backed up with reference and numbers with why it is particularly advantageous compared to other materials that have been explored.

Response: According to the reviewer's suggestions, we have added the references' comparison and explanation about the advantageous pyroelectric performance of Bi₂WO₆. As it is well known, the ferroelectric property of Bi₂WO₆ has been reported in many articles, it has a large spontaneous polarization (50 μC·cm⁻²) and a high Curie temperature of 950 K (*J. Phys. Chem. C* 2014, 118, 13514; *Phys. Rev. B* 2004, 70, 214112). The pyroelectric coefficient of bismuth tungstate at room temperature is 1.7 nC/cm²·K (*Ferroelectrics* 2002, 266, 259), which is roughly an order of magnitude with the previously-reported ZnO (*Nano Lett.* 2012, 12, 2833), but smaller than some traditional ferroelectric materials (*Energy Environ. Sci.* 2018, 11, 2198; *Nano Lett.* 2012, 12, 6408). Meanwhile, it has been reported that the pyroelectric performance is affected by the size of nanomaterials (*J. Appl. Phys.* 2010, 108, 042009). The pyroelectric coefficient strongly increases with the decreasing radius. Size-driven enhancement of pyroelectric coupling leads to giant pyroelectric current and voltage output of ferroelectric nanoparticles under temperature fluctuation (*Adv. Mater.* 2012, 24, 5357). Considering two-dimensional layer-structured Bi₂WO₆ nanoplates with relatively large specific surface, Bi₂WO₆ nanoplates is chosen as a typical pyroelectric material to testify the efficient pyroelectrocatalytic CO₂ reduction.

3. "More importantly, the suitable energy band structure and surface properties of Bi₂WO₆ allow it for CO₂ reduction" - How do the band structure and pyroelectric charge work together for CO₂ reduction? This is not clear; for example, it seems to have very little link to the final mechanism in Fig. 5.

Response: To clarify the process of pyroelectrocatalysis, the positive and negative charges are generated from catalyst by temperature variation, which participate in the similar electrochemical reactions of CO₂ reduction to the photocatalysis (*Chem. Rev.* 2019, 119, 6, 3962; *Nat. Commun.*

2020, 11, 1443). Here, the difference between pyroelectrocatalysis and photocatalysis is the excitation source. The pyroelectric charges are generated by temperature variation of pyroelectrics. When the pyroelectrics temperature remains a stable value, pyroelectrics are initially in equilibrium with bound surface charges due to the spontaneous polarization; Once heated, the polarisation of the pyroelectric is reduced so that some of the bound charge is free to take part in reduction-oxidation reactions; When the material is subsequently cooled, the polarisation of the material increases and charges move to the surface to balance the uncompensated screening charge carriers, leading to further reduction-oxidation reactions. During the pyroelectrocatalytic CO₂ reduction process, to satisfy the thermodynamic criteria, the reduction potential of electrons needs to be low enough. Different reduction products require different minimum potentials. For example, the electrons in conduction band should be lower than -0.38 V (vs. NHE) to generate CH₃OH in CO₂ reduction process. Many previous articles have similarly reported photocatalytic CO₂ reduction into CH₃OH using Bi₂WO₆ as catalyst (*Angew. Chem. Int. Ed.* 2015, 54, 13971; *J. Phys. Chem. C* 2016, 120, 18191; *Appl. Catal. B-Environ.* 2018, 238, 119). All address that the conduction band position of Bi₂WO₆ is suitable for CO₂ reduction to CH₃OH. Bi₂WO₆ {001} surface is a particular reactive surface energetically favoring the reduction of CO₂. Namely, CO₂ can be easily dissociated on the Bi₂WO₆ {001} surface (*ACS Appl. Mater. Interfaces* 2011, 3, 3594). More detail information can be seen on Pages 11 and 12 of theoretical calculation section in the manuscript.

4. “The voltage produced by pyroelectric effect can be a power source for electrochemical reaction. “Unusual terminology since voltage on its own to not a power - it is the potential driving force? Again this is not linked well to the modelling work or the final mechanism,

Response: According to the reviewer’s comments, we revised the link between the modelling work and the final mechanism. In addition, we corrected the terminology of power in the revised manuscript, that the voltage produced by pyroelectric effect can be a driving force for electrochemical reaction.

Pyroelectric materials' surface can generate voltage during the heating and cooling process. The pyro-potential U built in a pyroelectrocatalytic particle can be expressed as following equation,

$$U = (p \cdot \Delta T \cdot l) / \epsilon \quad (2)$$

where l , p , ΔT and ϵ are the size, the pyroelectric coefficient, the temperature changes and the permittivity of a pyroelectric particle, respectively (*Phys. Rev. B* 2016, 93, 195428). The pyro-potential distribution across a 2D Bi₂WO₆ nanoplate fitted by COMSOL finite element simulation is shown in **Fig. R2**, in which different colors represent different potentials. It can be seen potential difference occurs on the surfaces of the Bi₂WO₆ nanoplate.

Fig. R2 The COMSOL finite element simulation of pyro-potential of a single Bi₂WO₆ nanoplate.

Usually, the voltage output along the polarization direction can be vastly larger than the band gap of the pyroelectric material (*Phys. Chem. Chem. Phys.* 2014, 16,10408), which can be a driving force for electrochemical reactions (*Adv. Energy Mater.* 2019, 1902714). Ultimately, this driving force comes from pyroelectric effect through heating and cooling process. The reaction processes are shown in **Fig. R3**. Firstly, three H ions prefer to bind to the C atom, which breaks a C-O bond and results in a CH₃O* radical and a separate O ion (*step* “3”). Then the subsequent H ions will be attracted by the separated O ion until a H₂O molecule forms (*step* “5”). Finally, a methanol (CH₃OH) molecule is produced after one more H ion attaches to the CH₃O* radical (*step* “6”), the total equation can be expressed as following equation,

Fig. R3 Reaction processes for CO₂ reduction to CH₃OH.

5. For the pyroelectric current measurement, is there a polarization direction for each nanoplate since there is no poling process - this should be made clear and can be indicated with an arrow for the polarization direction (again would link better to Fig. 5).

Response: This is a good point. According to the previous report, the spontaneous polarization of Bi₂WO₆ is about 50 μC·cm⁻² (*J. Phys. Chem. C* 2014, 118, 13514), and we have added arrows in Fig. 5 of our revised manuscript to clearly indicate the direction of spontaneous polarization.

All ferroelectrics are pyroelectric, where the non-centrosymmetry plays a vital role in the

piezo-/pyroelectric effect. For bulk ferroelectric polycrystalline ceramics, poling is necessary to align the randomly oriented domains in order to obtain a macroscopic piezo-/pyroelectric effect. While these chemically-synthesized piezo-/pyroelectric micro/nanomaterial particles often grow along a certain crystallographic direction and can possess the crystal anisotropy and spontaneous ferroelectric polarization (*Nat. Commun.* 2010, 1, 93; *Appl. Phys. Lett.* 2006, 89, 263119; *Chem. Soc. Rev.* 2017, 46, 7757). Therefore, those nanoparticles usually behave the pyro-/piezoelectric effect without a pre-poling process (*Appl. Phys. Lett.* 2016, 109, 032904; *J. Am. Ceram. Soc.* 2011, 94, 3812; *Chem. Commun.* 2013, 49, 4003). Yu *et al.* reported that the piezoelectric constant d_{33} of BaTiO₃-based nanomaterials could be up to 13 pC/N (*Appl. Phys. Lett.* 2016, 109, 032904). Wang *et al.* reported that the d_{33} of BaTiO₃ nanowires could reach 45 pC/N (*Nano. Letts.* 2007, 7, 2966). In addition, it has been reported that some graphene-like two dimensional monolayer materials, such as MoS₂ can also exhibit piezoelectric effect due to the broken centro-symmetry (*Nature* 2014, 514, 470).

6. There thermal cycling is between 15 and 70 degC. What is the exact change in polarisation of the material in this range (C/m²)? This is important as it provides the charge for the process. What would then be the voltage generated by such charge across a particle?

Response: The pyroelectric coefficient of Bi₂WO₆ is 1.7 nC/cm²·K (*Ferroelectrics* 2002, 266, 259), accordingly, the change in polarization of Bi₂WO₆ between 15 °C and 70 °C is about 93.5 nC/cm². When the temperature changes, the polarization intensities of pyroelectric materials will change, and the balance between polarization and shielding charge is disturbed, so the free charges will be generated. Since Bi₂WO₆ is a wide bandgap semiconductor ($E_g = 2.69$ eV) (*ACS Appl. Mater. Interfaces* 2011, 3, 3594), which can be regarded as a capacitor. According to the following formula,

$$C = Q/U \quad (4)$$

where C , Q and U are capacitance, charge and voltage, respectively. The pyroelectric-induced charges on the Bi₂WO₆ surface can generate an electric field, resulting in a potential difference along the polarization direction.

7. A stronger link between the simple mechanism of Fig. 5 and the modelling in Fig. 4 (for example - some additional aspects of Fig. 4 could be added to Fig. 5 since the final mechanism is rather simplistic).

Response: In view of the reviewer's opinion, we have enriched the catalytic mechanism in Fig. 5 of our revised manuscript to better link the mechanism and the modelling. In Fig. 4, we calculate every reaction energy of CO₂ on the Bi₂WO₆ surface to demonstrate the possibility of CO₂ reduced to CH₃OH. In addition, we have redrawn Fig.5 to more clearly present the pyroelectric process in the CO₂ reduction process.

8. Experimental queries: Was the materials suspended or does it fall to the bottom of the solution. For the thermal cycling it is stated that "The sample is then subjected to alternating temperature-variation cycles", this could be described more detail.

Response: In view of the reviewer's comments, we have made corresponding changes to the relevant content: High purity CO₂ gas was bubbled into the borosilicate tube for 10 min. Then the tube was immediately sealed with a rubber stopper. The sample was suspended in the solution under stirring, being applied alternating temperature between 15 °C and 70 °C in water bath. The entire catalytic process is performed in dark.

The relevant discussions have been added in Pages 17 of our revised manuscript.

Second round of Reviewer #1

I am happy with the corrections in response to reviewer comments and the paper is improved.

Response: Thanks.

First round of Reviewer #2

The manuscript by Xiao *et al.* reports the use of Bi_2WO_6 nanoplates as pyroelectrocatalyst that converts CO_2 into methanol by harvesting the energy from temperature variation below $100\text{ }^\circ\text{C}$. The results showed that this system could operate with high energy efficiency after 20 cycles of temperature variation. DFT calculations were performed to better understand the catalytic mechanism of CO_2 reduction. Even though the authors claimed that they have presented a novel method for efficient conversion of CO_2 to methanol, there are many points deserving clarification and which cannot suggest publishing in Nature Communication journal.

Response: Thank the reviewer for the positive and constructive comments. As the reviewer said, our pyroelectrocatalyst converts CO_2 into methanol by harvesting the energy from temperature variation below $100\text{ }^\circ\text{C}$. The results showed that this system could operate with high energy efficiency after 20 cycles of temperature variation. Density functional theory (DFT) calculations were performed to better understand the catalytic mechanism of CO_2 reduction. According to the reviewer's suggestion, we have made specifically point-to-point revisions and clarifications to meet the reviewer's requirements.

1. First, the authors didn't discuss possible formation of other CO_2 reduction products in their experiments. Is methanol the only product of that reaction? What is the selectivity for CH_3OH formation?

Response: According to the reviewer's suggestion, we have conducted a comprehensive examination of the reduced products after 20 thermal cycles. For the detection of reaction products in solution, $800\text{ }\mu\text{L}$ reaction solution, $100\text{ }\mu\text{L}$ D_2O and $10\text{ }\mu\text{L}$ DMSO (0.1% vol aqueous solution) were taken into nuclear magnetic tubes, and detected with nuclear magnetic resonance (NMR) spectrometer with superconducting magnet (AVANCE NEO 400MHz, Switzerland). Here the added DMSO is used as the internal standard reference. From the ^1H NMR spectra shown in **Fig. R1**, only methanol can be detected in liquid phase.

Fig. R1 ^1H NMR spectra of the pyroelectrocatalytic reaction solution.

The gaseous products are also analyzed by a gas chromatograph (7890B, USA) through manual injections using a thermal conductivity detector with N₂ as the carrier gas. There is only a small amount of CH₄ and CO in the gaseous product, as seen in **Fig. R2** (0.11 μmol·g⁻¹ and 0.20 μmol·g⁻¹, respectively), indicating high selectivity of Bi₂WO₆ pyroelectrocatalytic CO₂ reduction to CH₃OH.

Fig. R2 Gas products of the pyroelectrocatalysis.

The relevant discussions have been added in Pages 8 of our revised manuscript.

2. The authors discuss the formation of OH radical as an important intermediate in the formation of CH₃OH. However, the discussion of the actual CO₂ reduction reaction is extremely limited. This process requires 6 electrons and therefore cannot be performed in a direct one-step reaction once the OH radicals have been generated.

Response: As pointed out by the reviewer, the CO₂ reduction reaction process requires 6 electrons. The reaction processes are shown in **Fig. R3**. Firstly, three H ions prefer to bind to the C atom, which breaks a C-O bond and results in a CH₃O* radical and a separate O ion (*step "3"*). Then the subsequent H ions will be attracted by the separated O ion until a H₂O molecule forms (*step "5"*). Finally, a methanol (CH₃OH) molecule is produced after one more H ion attaches to the CH₃O* radical (*step "6"*), the total equation can be expressed as following equation,

Fig. R3 Reaction processes for CO₂ reduction to CH₃OH.

In fact, the ·OH radicals are not the intermediate in the formation of CH₃OH. The detection of ·OH radicals illustrates the redox reactivity of positive and negative electric charges generated under temperature variation. In our process, pyroelectric induced positive and negative electric charges can react with O₂ and OH⁻, with the corresponding redox potentials -0.33 V (vs. NHE) and 2.38 V (vs. NHE) (*ACS Appl. Mater. Interfaces* 2017, 9, 2899), respectively. The reaction product O₂⁻ and ·OH are the main active groups of catalytic degradation of dyes (*Sci. Rep.* 2020, 10, 1). The maximum reduction potential of CH₃OH is -0.38 V (vs. NHE), which is proximate with the maximum reduction potential of OH⁻ (-0.33 V vs. NHE), indicating the pyroelectrocatalytic activity for CO₂ reduction.

3. A very important point for such a catalytic paper would be the labeling test of ¹³CO₂, which however is missing in this manuscript.

Response: According to the reviewer's suggestion, we have added the isotopic labeling experiment using ¹³CO₂ as feedstock to prove the methanol is coming from CO₂ reduction. The ¹H NMR spectrum of the reaction solution in **Fig. R4a** clearly shows the formation of CH₃OH (δ = 3.34 ppm) when the unlabeled CO₂ is used as feedstock. When using ¹³CO₂ instead of CO₂, the ¹H NMR spectrum of the reaction solution is shown in **Fig. R4b**. The double peaks between 3.7 and 3 ppm can be attributed to the proton coupled with the ¹³C in ¹³CH₃OH (*J. Am. Chem. Soc.* 2013, 135, 4596). The small peak of CH₃OH at 3.34 ppm possibly comes from the decomposed CO₂ of NaHCO₃ during the heating process, participating in CO₂ reduction (*J. Am. Chem. Soc.* 2014, 136, 1734). The results testify that CO₂ is the carbon source for the pyroelectrocatalytic CO₂ reduction into CH₃OH.

Fig. R4 ¹H NMR spectra of the pyroelectrocatalytic reaction solution (a) with the unlabeled CO₂ (b) with the labeled ¹³CO₂.

The relevant discussions have been added in Pages 10 and 11 of our revised manuscript.

4. The computational part discusses the addition of H ion, while it is generally accepted to have a

concerted proton-electron ($H^+ + e^-$) steps, making use of the computational hydrogen electrode approach. Currently, it is not clear how the authors took into account the reduction steps.

Response: Intuitively, the hydrogen of the reaction in solvent exists as H^+ . However, in quantum mechanics and DFT, the electron is not a point charge but spreads in a certain region as electron cloud which is determined by its wavefunction. In DFT calculations, we deal with the wavefunctions of electrons in a specific potential originated from the nuclear charges of atoms. Therefore, it is impossible to separate hydrogen atom as proton and electron directly in DFT calculation. To model the reaction between H and the radical on the Bi_2WO_6 surface, we place a H atom beside a certain site of the radical and carried out DFT calculations to optimize the interaction between them. Electron charge transfer happens between the H atom and radical, usually from H to the radical so that the H atom finally becomes H^+ , according to the chemical bonding between H and the radical (e.g. CO_2 molecule in this work). In other words, charge separation can be reached after self-consistent-field iterations.

The relevant discussions have been added in Pages 12 of our revised manuscript.

5. Are these gas-phase calculations or is the water solvent explicitly considered? The authors say in the text that "The CO_2 reduction starts when the hydrogen ions in the solvent interact with the CO_2 molecule." This is not reflected in Fig. 4.

Response: The calculations are based on gas phase. The solvent here carries hydrogen ions but does not involve in the reaction of CO_2 reduction. Therefore, it is not necessary to consider the water solvent in DFT calculations. In addition, we usually investigate the reaction between an individual H atom and an individual radical in DFT calculations, so it will make the calculations much more complex if we include the water solvent, and this is computationally unaffordable. In Fig. 4b, we only present the final structural configuration of each step, i.e. the process of "interaction" has been done in DFT calculation through optimizing the C-H or O-H bond length.

The relevant discussions and more explanation have been added in Pages 12 and 13 of our revised manuscript.

6. How the water formed near the methanol molecule favors its release? How eventual release of methanol at steps 4 or 5 suggested by the authors and rejected based on higher activation energies possibly explain the fact that 6 electron reductions have not yet been reached?

Response: Water molecule can form at *step "5"* but is not released from the Bi_2WO_6 surface because its release requires an activation energy of about 1.7 eV. Meanwhile, the formation of methanol at *step "6"* requires only about 0.2 eV. Therefore, the formation of methanol is much favorable than the release of water at *step "6"*. Nevertheless, the activation energies for the water and methanol release at *step "7"* are comparable. More explicitly, the activation energy for water release is about 0.15 eV lower than that of methanol release (1.6 eV). We have added more discussion about this in Pages 12 of the revised manuscript.

At each step of calculations, we considered all possible products, where the H atom binds to the radical at different sites, and then compared the corresponding reaction energies. In Fig. 4b, we only plotted the atomic structure of the lowest-energy product or most possible product at each step. At *step* “4” or “5”, there is possible product of methanol if the H atom binds to the CH_3O^* radical rather than the O^* or OH^* radical. However, the corresponding energy is much higher than the main product. Accordingly, the production rates of methanol at *step* “4” or “5” is ignorable.

Second round of Reviewer #2

The authors have significantly improved their manuscript taking into account all questions and concerns raised by the referees. I suggest publication after polishing some repetitions and improving the English (for instance, page 13 lines 269-273 and page 14 lines 291-294)

Response: Thanks for the reviewer's comments. We have improved the English in the revised manuscript.

First round of Reviewer #3

This paper describes detailed characterization of Bi_2WO_6 using XRD, SEM, TEM, HR-TEM, HAADF-STEM, Piezoelectric Force microscopy, and pyrocurrent tests. The sample is tested for CO_2 reduction using Na_2SO_3 as reducing agent and also rhodamine B decomposition tests were also performed. The characterizations would attract readers, e.g, very fine HAADF-STEM images in comparison crystal structure model (Fig. 1f), however, it would not be novel point of this paper. The CO_2 reduction tests are very tentative, and support data including tests in the absence of Na_2SO_3 , radical trapping tests, dye decomposition tests, and DFT calculations are totally out of focus to justify the CO_2 pyroreaction data (Fig. 3a and b). The reasons are described in the comments below. In my opinion, this paper does not satisfy the criteria of reliable CO_2 photo and/or pyroreduction paper.

Response: Thank the reviewer for the positive and constructive suggestions. As the reviewer's statement, we present a detailed characterization of Bi_2WO_6 using XRD, SEM, TEM, HR-TEM, HAADF-STEM, piezoelectric force microscopy, and pyrocurrent tests. The characterizations would attract readers, e.g. very fine HAADF-STEM images in comparison crystal structure model. In addition, we have carried out the direct evidence for justifying the pyroelectrocatalytic CO_2 reduction to methanol. For example, experiments with or without Bi_2WO_6 nanoplates at different temperature at 15 °C, 45 °C and 70 °C have been performed (**Fig.R1**), in which only methanol can be produced with the presence of both Bi_2WO_6 nanoplates and temperature variation. In addition, ^{13}C labeled test further proved that the carbon in methanol came from CO_2 . All of these experiments confirm the fact of pyroelectrocatalytic CO_2 reduction to methanol. We believe that our revised manuscript satisfies the criteria of reliable pyroelectrocatalytic CO_2 reduction paper.

Fig. R1 (a) pyroelectrocatalytic CO_2 conversion without Bi_2WO_6 (b) pyroelectrocatalytic CO_2 conversion at different stable temperature.

1. Page 2, line number 41: "biologic" should be "biological".

Response: The relevant content has been modified in the revised manuscript.

2. Page 2, line numbers 43–44: The meaning of phrase is unclear: "the use of harsh/expensive reaction conditions and/or low efficiency". What is used in such cases?

Response: To make the meaning of phrase clear, we have rewritten the two lines as followings: hydrogenation of CO₂ to form CH₃OH process requires high operating temperatures (200–250 °C) and high pressures (5–10 MPa), which limits the yield of methanol (*ChemSusChem* 2016, 9, 322; *Fuel* 2008, 87, 443). Photocatalytic reduction of CO₂ can be carried out in relatively mild temperature and pressure, but it always suffers drawbacks, such as insufficient light absorption and no response under dark condition (*Adv. Mater.* 2014, 26, 4607; *Appl. Catal. B-Environ.* 2019, 258, 117957). Therefore, it is imperative to develop alternative, cost-effective, and environmentally friendly approach for CO₂ conversion.

The relevant discussions have been added in Pages 2 of our revised manuscript.

3. Page 2, last line: "to develop" is repeated. The authors need to check their manuscript.

Response: The sentence has been revised in the revised manuscript.

4. Page 4, last sentence in the Introduction section: "far" is not used for temperature. Furthermore, the grammar needs to be checked in various parts in the Introduction section.

Response: The relevant content has been carefully modified in the revised manuscript.

5. Page 6, line number 119: It is weird to mention "microscope was performed".

Response: The description has been modified in the revised manuscript.

6. Page 9, line number 190: The wavelength 650 nm is confusing. In Fig. 3d, intense, broad peak at 710 nm accompanies a shoulder peak at 630 nm. If one of them is due to O₂⁻ radical anion, what was the source for the other peak?

Response: In this work, O₂⁻ was detected by Nitro Blue Tetrazolium (NBT) inspection agent, which is a widely used chemical compound for the determination of superoxide anion radical (O₂⁻) and superoxide dismutase activity in many fundamental biological processes. Its reduction reaction with superoxide and the associated spectral absorption data of monoformazan (MF) and diformazan (DF) (*Ind. Eng. Chem. Res.* 2009, 48, 9331). The reaction process can be described as following equations,

The absorption peak of MF and DF are near 630 nm and 720 nm, respectively.

The relevant discussions have been added in Pages 9 and 10 of our revised manuscript.

7. Page 9, line number 194: The equations 5 and 6 are not found in main text. The description would be "equations 2-4. However, they are the reaction of O₂ or hydroxy reduction. The

reduction potential of CO₂ is different from these, and the equations 3 and 4 are not directly related to the reduction of CO₂.

Response: On basis of the review's suggestion, we have revised the equation numbers and made a detail clarification on these equations.

In our process, pyroelectric induced positive and negative electric charges can react with O₂ and OH⁻, with the corresponding redox potentials -0.33 V (vs. NHE) and 2.38 V (vs. NHE) (*ACS Appl. Mater. Interfaces* 2017, 9, 2899), respectively. The reaction product O₂⁻ and ·OH are the main active groups of catalytic degradation of dyes (*Sci. Rep.* 2020, 10, 1). The maximum reduction potential of CH₃OH is -0.38 V (vs. NHE), which is proximate with the maximum reduction potential of OH⁻ (-0.33 V vs. NHE). Our pyroelectrocatalytic material has been demonstrated the ability for CO₂ reduction.

The relevant discussions have been added in Pages 9 of our revised manuscript.

8. Page 9-10: I cannot understand why the photocomposition (Noted: it should be pyrocomposition) tests were performed for rhodamine B. It is totally unrelated to pyroreduction of CO₂.

Response: The misunderstanding might confuse the reviewer. Except for the pyroelectrocatalysis of CO₂ reduction, we further performed RhB pyroelectrocatalytic decomposition experiment to fully demonstrate the pyroelectrocatalytic activities. In fact, RhB pyroelectrocatalytic decomposition is a visual evidence to prove the redox ability of pyroelectric generated charges.

In order to avoid confusion of CO₂ reduction and RhB decomposition, we have moved the related contents into the supporting information.

9. Page 12, lines number 261-262: It is extremely questionable to conclude that the CO₂ reduction was "exothermic" and to mention "spontaneous" because the formation enthalpy of CO₂ is very negative (-394 kJ/mol).

Response: As the reviewer said, the formation enthalpy of CO₂ is very negative. The binding energy on the Bi₂WO₆ surface is -3.6 eV per molecule in our calculation. Nevertheless, the interaction between H and CO₂ is still very strong. As can be seen in **Fig. R2**, the first four reaction steps are exothermic with large negative reaction energies. Even through the last three reaction steps are endothermic, the overall process of the CO₂ reduction is still exothermic as shown in **Fig. R2**. Therefore, the activation energies required in the last three reaction steps can be compensated by the energy released in the first four steps. Therefore, the CO₂ reduction lasts spontaneously in principle. However, given the energy loss in a solvent environment, energy supply is still needed but the energy demanding is not much. This is reason that the temperature is not high in our experiments.

We have rewritten these sentences on Page 13 to explain this point.

Fig. R2 Reaction energies for the CO₂ reduction. Eight reaction steps are considered. The reaction energies for the side product (methanol) at steps “4” and “5” are indicated by short dashed lines. The short red (dashed) lines are estimated from the free energies.

10. Fig. 5 and page 13: What happens in the absence of Na₂SO₃ (Fig. 3a)? Was O₂ formation confirmed? Furthermore, ¹³CO₂-labeled confirmation test is also essential.

Response: To solve the problems of carriers’ recombination, sacrificial electron donors have been frequently added into the reaction system to consume the generated holes, thereby increasing the survival time of generated electrons. It includes two steps: Firstly, electric charge carriers are generated under light or thermal signals *via* photovoltaic or pyroelectric effect, respectively. Secondly, the generated positive and negative electric carriers induce the chemical oxidation-reduction reactions. The position of the valence band of Bi₂WO₆ conditions satisfy for oxygen production (*Appl. Catal. B-Environ.* 2017, 219, 209; *Appl. Catal. B-Environ.* 2018, 238, 119). It has been similarly reported that CO₂ reduction using Bi₂WO₆ as catalyst generated oxygen when there is no sacrificial agent added. Kakekhani *et al.* have theoretically proved that the pyroelectrocatalytic product of water splitting includes not only H₂, but also O₂ (*J. Mater. Chem. A* 2016, 4, 5235). Therefore, oxygen will be produced in the pyroelectrocatalytic process without the addition of sacrificial agents.

According to the reviewer’s suggestion, we added the isotopic labeling experiment using ¹³CO₂ as feedstock in order to prove that CH₃OH is the product of CO₂ reduction. The ¹H NMR spectrum of the reaction solution in **Fig. 3a** clearly showed the formation of CH₃OH ($\delta = 3.34$ ppm) when the unlabeled CO₂ was used as feedstock. **Fig. 3b** shows the ¹H NMR spectrum of the reaction solution when using ¹³CO₂ instead of CO₂. The double peaks between 3.7 and 3 ppm can be attributed to the proton coupled with the ¹³C in ¹³CH₃OH (*J. Am. Chem. Soc.* 2013, 135, 4596). The small peak of CH₃OH at 3.34 ppm possibly comes from the decomposed CO₂ of NaHCO₃ in

the reaction solution, participating in CO₂ reduction during the heating process (*J. Am. Chem. Soc.* 2014, 136, 1734). The results testify that CO₂ is the carbon source for the pyroelectrocatalytic CO₂ reduction into CH₃OH.

Fig. R3 ¹H NMR spectra of the pyroelectrocatalytic reaction solution (a) with the unlabeled CO₂ (b) with the labeled ¹³CO₂.

The relevant discussions have been added in Pages 10 and 11 of our revised manuscript.

Second Round of Reviewer #3

This paper reports catalysis when the temperature change cycle of catalyst was repeated. The data is only one repetition condition, and is not enough to prove pyro-originated catalysis. Directly related characterization is lacking.

Response: According to the reviewer's comment, we have strengthened the experimental proof (temperature derivation including the rate and the variation of temperature range) on the pyro-originated catalysis in the revised manuscript. The details are described in the following response (3). In addition, we have added and provided direct characterizations to support the process of pyroelectrocatalysis in the revised manuscript. Piezoelectric force microscope characterization indicates the ferroelectricity of Bi₂WO₆ (Fig. 2a-d), Pyroelectric current measurement (Fig. 2e-f), free radical detections (Fig. 3c-d) and ESR characterization (Fig. S8) prove the generation and separation of pyroelectric charges. Isotope labeling experiments further demonstrate the occurrence of pyroelectrocatalytic CO₂ reduction to methanol (Fig. 3e-f). Comparative experiments on pyroelectrocatalysis in Fig.3a-b and Fig. S7 shows the CO₂ reduction reaction can only occur when both the temperature variation and the catalyst are present at the same time. Density functional theory calculations have also been performed to better understand the pyroelectrocatalytic mechanism of CO₂ reduction (Fig. 4). All these experimental results and theoretical calculations have confirmed the pyro-originated pyroelectrocatalytic CO₂ reduction of Bi₂WO₆ under temperature variation.

(1) Page 3, lines 1-2: It is not fair comparison. Photocatalysts can be fully irradiated under light. It is not always rainy days on the earth.

Response: Photocatalysis, as an advanced method, can be carried out in mild temperature and pressure and have attracted much attention due to its potential application in converting CO₂ into hydrocarbon fuel. The use of clean and renewable solar energy changing CO₂ into hydrocarbon fuel by photocatalysis can not only achieve the purpose of energy saving and suppression of the greenhouse effect, but also alleviate the problem of energy shortage. As a matter of fact, photocatalytic reduction of CO₂ can be carried out in mild temperature and pressure, but it does not work in dark (*Chem. Soc. Rev.* 2020, 49, 2937; *Energy Environ. Sci.* 2018, 11, 2198). As for pyroelectrocatalysis, temperature variation in daily life is used for clean energy in our work. It would be desirable to utilize the natural temperature variation for CO₂ conversion. This article introduces a new method of CO₂ reduction through temperature variation even in dark.

Taking the reviewer's suggestion into account, the relevant content in Page 3, lines 1-2 of the manuscript has been rewritten to make the readers easy to understand the difference between photocatalysis and pyroelectrocatalysis in utilizing the environmental clean solar light energy and the cold-hot alternation thermal energy.

(2) Page 3, line 3 from the bottom: The language is not improved. It should be "The study has been reported for seventy years." Page 7, line 10: "can generation" would be "can generate".

Response: Thanks for the reviewer's comments. We have carefully edited the whole manuscript to enhance English.

(3) The catalytic result section: Rather than to demonstrate both CO₂ and Rhodamine B decompositions, it is essential to prove the temperature derivation (the rate and frequency, the variation of temperature range) was the origin of the catalysis.

Response: Thanks for the reviewer's comments. Temperature derivation (the rate and frequency, the variation of temperature range) is very important influence factors on pyroelectrocatalysis.

In general, a high rate of temperature change ($\partial T/\partial t$) is beneficial for pyroelectricity. It has been reported that the pyroelectric charge is proportional to $\partial T/\partial t$ in the range of 10⁻² to 10⁶ °C/s (*J. Appl. Phys* 2012, 112, 104106). Here, we have investigated the effect of temperature change rate on the pyroelectrocatalytic degradation dyes in **Fig. R1**, indicating the more rapid speed of degradation along with the higher values of $\partial T/\partial t$.

Fig. R1 Pyroelectrocatalytic dye degradation under different temperature change rates.

The range of temperature can also affect the pyroelectrocatalytic CO₂ reduction. **Fig. R2** shows the yield of methanol after 10 thermal cycles in different temperature ranges (15-40 °C, 15-50 °C, 15-70 °C, 15-85 °C). In **Fig. R2**, the methanol yield increases as the temperature range increases. The pyro-induced charges (dQ) can be expressed in Eq. (1)

$$dQ = p \cdot A \cdot dT \quad (1)$$

where p is the pyroelectric coefficient of Bi₂WO₆. A is the Bi₂WO₆-coated electrode area. dT is the temperature range. The large range of temperature can produce more pyrocharges, resulting in the better pyroelectrocatalytic performance. The relevant discussions have been added to the revised manuscript.

Fig. R2 Methanol yield of different temperature ranges.

(4) Figure 3: Why was a half of ¹²CH₃OH observed in the ¹³CO₂-labeled test? Just one point chart in the reaction test is insufficient.

Response: In the catalytic CO₂ reduction systems, NaHCO₃ or KHCO₃ is often used as a buffer (*J. Am. Chem. Soc.* 2017, 139, 16161; *J. Am. Chem. Soc.* 2014, 136, 1734; *Adv. Mater.* 2019, 31, 1804710; *Adv. Mater.* 2019, 31, 1804710). The ¹²CO₂ is generated from the heating of HCO₃⁻ ions. Therefore, both ¹³CO₂ and ¹²CO₂ participate in the catalytic reactions to form methanol in the ¹³C

isotope experiment, which was also observed in the $^{13}\text{CO}_2$ -labeled test (*Joule* 2018, 2, 1369). To avoid $^{12}\text{CO}_2$, we have completed the additional experiment using NaOH instead of NaHCO_3 . **Fig. R3** shows the NMR spectra of the pyroelectrocatalytic reaction solution with labeled $^{13}\text{CO}_2$ using NaOH as buffer. There is only $^{13}\text{CH}_3\text{OH}$ signal in **Fig. R3**, confirming that the methanol originates from the pyroelectrocatalytic CO_2 reduction. In addition, the more points' chart has been presented in the Fig.S3 of supporting information. The relevant discussions have been added to the revised manuscript.

Fig. R3 ^1H NMR spectra of the pyroelectrocatalytic reaction solution with labeled $^{13}\text{CO}_2$.

(5) The extent of pyro-derived charge separation, e.g. in Figure 5, is not experimentally demonstrated in this paper.

Response: We have presented the experiments to demonstrate the pyro-derived charge separation (**Fig. R4 and 5**). **Fig. R4** shows that $\cdot\text{OH}$ and O_2^- are detected by fluorescence spectrometry using terephthalic acid as a photoluminescent $\cdot\text{OH}$ trapping agent and by UV-Vis spectrophotometer using nitro-blue tetrazolium as absorption O_2^- trapping agent, respectively.

Fig. R4 (a) The fluorescence spectra of 2-hydroxyterephthalic acid, (b) The absorption spectra of diformazan and monoformazan.

We have further measured the electron spin resonance (ESR) signals of radical's spin-trapped by

5,5-dimethyl-1-pyrroline-N-oxide to verify the generation of the $\cdot\text{OH}$ and $\text{O}_2^{\cdot-}$. **Fig. R5** shows the characteristic peaks of DMPO- $\cdot\text{OH}$ and DMPO- $\text{O}_2^{\cdot-}$ spin adducts with the increase of thermal cycles, indicating the amount increase of both $\cdot\text{OH}$ and $\text{O}_2^{\cdot-}$. In contrast, there is no radical signals without the thermal cycles. Moreover, the pyro-current response in Fig. 2f-g can also demonstrate the pyro-derived charge separation.

Fig. R5 ESR spectra of (a) DMPO- $\cdot\text{OH}$ and (b) DMPO- $\text{O}_2^{\cdot-}$ adducts after undergoing different thermal cycles.

As it is well known, pyroelectric materials can generate the positive and negative charges under cold-hot temperature variation (*Nat. Mater.* 2018, 17, 432). The temperature change of the pyroelectric material causes the imbalance between its spontaneously polarization and surface charges, which results in the generation of free charge. The pyro-induced positive and negative charges can react with O_2 and OH^- in water to form $\text{O}_2^{\cdot-}$ and $\cdot\text{OH}$, respectively, as shown in Eq. (2)-(4).

The relevant discussions have been added to our revised manuscript.

Third Round of Reviewer #3

This paper reports the EM characterization of layered Bi_2WO_6 and the feasibility of pyrocatalysis for CO_2 reduction. This reviewer does not recommend this paper for publication due to following reasons.

(1) Pages 1-7: The language is still difficult to read. Minor errors still remain, e.g. Fig. 2e instead of Fig. 2f (page 7, line 6 from the bottom) and Fig. 2f instead of Fig. 2g (page 7, line 5 from the bottom).

Response: Thanks for reviewer's comment. The relevant content has been carefully modified and highlighted in the revised manuscript.

(2) Pages 8-11: Product O₂ is not monitored even qualitatively. It is doubtful why methanol was always happily produced selectively. The impurity transformation needs to be carefully checked. The same opinion is repeated. One-point check of isotope-label test in the time course of pyrocatalysis is insufficient because methanol is often formed starting from impurity, and time course monitoring is essential to prove the reaction

Response: According to the reviewer's suggestion, we have added O₂ monitor experiments (Figure R1) and many thermal cycles of isotope-label test in the time course of pyroelectrocatalysis (Figure R2). Our experimental results verify the CH₃OH is totally from CO₂ reduction not from any impurity. Usually, organic impurities in carbon-based catalysts can possibly be oxidized to hydrocarbons (*Nat. Commun* 2020, 11, 1). Nevertheless, our catalyst is inorganic Bi₂WO₆ without carbon. As for the whole reaction process, we have carefully checked and carried out each step experiment and test. Oxygen as a byproduct of pyroelectrocatalytic CO₂ reduction is measured by gas chromatography (Fuli GC9790plus). Fig. R1 shows the amount of O₂ increases with the number of thermal cycles. The amount of oxygen are roughly proportional to the amount of methanol.

Fig. R1 The yield of O₂ with different thermal cycles.

The time dependence isotope-label ¹³CO₂ test is shown in Fig. R2. Before the reaction starts, neither ¹²CH₃OH nor ¹³CH₃OH are detected. Then, the yield of methanol increases as the increase

of thermal cycles, moreover, the signal of $^{12}\text{CH}_3\text{OH}$ is not detected throughout the whole catalytic reaction. Here, we have strictly controlled the carbon source in the catalysis process, no other carbon source is added except $^{13}\text{CO}_2$. The above data shows that methanol is not generated from impurities but from CO_2 reduction.

Fig. R2 NMR of isotope-label test with cumulative thermal cycling over time.

There are many reports regarding methanol to be produced from CO_2 through different catalytic processes (*Nat. Commun.* 2020, 11, 2531; *Angew. Chem. Int. Ed.* 2019, 58, 3804; *Chem. Rev.* 2019, 119, 3962–4179; *Joule* 2018, 2, 1369). As it is well known, the enthalpy change (ΔH) of CO_2 to methane (890 kJ/mol) is higher than that of methanol (727 kJ/mol) (*J. Mater. Chem. A*, 2018, 6, 22411). Meanwhile, molecular absorption and desorption have a great impact on the selectivity of catalysis (*ACS Catal.* 2020, 10, 8632; *Nano Energy* 2020, 75, 104959). As for Bi_2WO_6 , ferroelectric polarization can affect molecular adsorption and desorption from the surface of the materials (*J. Mater. Chem. A* 2016, 4, 5235). Moreover, it has been reported that the dissociated $\text{CO} + \text{O}$ on the Bi_2WO_6 surface is easily recombined to CO_2 (*ACS Appl. Mater. Interfaces* 2011, 3, 3594). All these points suggest methanol selectivity in our pyroelectrocatalytic process.